palaeontology/taxonomy and systematics/evolution

Chengjiangocarididae, Deuteropoda, Euarthropoda, Fuxianhuiida, *Lihuacaris ferox*, Radiodonta

**Authors for correspondence:**
Javier Ortega-Hernández
e-mail: jortegahernandez@fas.harvard.edu
Xi-guang Zhang
e-mail: xgzhang@ynu.edu.cn

# New multipodomerous appendages of stem-group euarthropods from the Cambrian (Stage 4) Guanshan Konservat-Lagerstätte

De-guang Jiao[1,2,†], Stephen Pates[3,4,†], Rudy Lerosey-Aubril[3], Javier Ortega-Hernández[3], Jie Yang[2], Tian Lan[5] and Xi-guang Zhang[2]

[1]Yuxi Normal University, Kunming, 134 Phoenix Road, Yuxi, Yunnan 653100, People's Republic of China
[2]Key Laboratory for Palaeobiology, Yunnan University, Kunming 650091, People's Republic of China
[3]Museum of Comparative Zoology and Department of Organismic and Evolutionary Biology, Harvard University, Cambridge, MA 02138, USA
[4]Department of Zoology, University of Cambridge, Cambridge CB2 3EJ, UK
[5]College of Resources and Environmental Engineering, Guizhou University, Guiyang 550003, People's Republic of China

SP, 0000-0001-8063-9469; RL, 0000-0003-2256-1872;
JO-H, 0000-0002-6801-7373; TL, 0000-0001-7713-5411;
X-gZ, 0000-0003-4697-0531

Stem-group euarthropods are important for understanding the early evolutionary and ecological history of the most species-rich animal phylum on Earth. Of particular interest are fossil taxa that occupy a phylogenetic position immediately crownwards of radiodonts, for this part of the euarthropod tree is associated with the appearance of several morphological features that characterize extant members of the group. Here, we report two new euarthropods from the Cambrian Stage 4 Guanshan Biota of South China. The fuxianhuiid *Alacaris*? sp. is represented by isolated appendages composed of a gnathobasic protopodite and an endite-bearing endopod of at least 20 podomeres. This material represents the youngest occurrence of the family Chengjiangocarididae, and its first record outside the Chengjiang and Xiaoshiba biotas. We also describe *Lihuacaris ferox* gen. et sp. nov. based on well-preserved and robust isolated appendages. *Lihuacaris ferox* exhibits an atypical combination of characters including an enlarged rectangular base, 11 endite-bearing podomeres and a hypertrophied distal element bearing 8–10 curved spines. *Alacaris*? sp. appendages display adaptations for macrophagy. *Lihuacaris ferox*

†Joint first authors

appendages resemble the frontal appendages of radiodonts, as well as the post-oral endopods of chengjiangocaridid fuxianhuiids and other deuteropods with well-documented raptorial/predatory habits. *Lihuacaris ferox* contributes towards the record of endemic biodiversity in the Guanshan Biota.

# 1. Introduction

Euarthropods, a major group whose extant representatives include chelicerates (e.g. horseshoe crabs and arachnids), myriapods (e.g. millipedes and centipedes) and pancrustaceans (e.g. insects and crustaceans), comprise over 80% of animal biodiversity today [1]. The fossil record demonstrates that euarthropods have been a significant faunal component of the Phanerozoic biosphere and key players in marine ecosystems since the Cambrian Period [2]. Much of our understanding of euarthropod origins and early evolution comes from exceptional fossil deposits that preserve non-biomineralized tissues, such as legs, eyes and guts. Exceptional fossils illuminate our understanding of the relationships of modern euarthropod groups [3] and also reveal the polarity of characters that accompanied the evolution from soft-bodied and annulated lobopodian ancestors to sclerotized segmented crown-group euarthropods [2,4–10]. Among the key characters that are recognizable among the so-called lower stem-group euarthropods (*sensu* Ortega-Hernández [6]) include the presence of paired serially repeating digestive glands, exemplified by the large-bodied lobopodians *Jianshanopodia* and *Megadictyon* [8], and swimming flaps in the more derived 'gilled lobopodians' *Kerygmachela* and *Pambdelurion* (e.g. [11–14]). Furthermore, the presence of arthropodization and well-developed compound eyes, some of the most recognizable morphological features of modern representatives of Euarthropoda, also appeared within the stem-lineage, particularly among the radiodonts [5,15].

Although the composition of lower stem-group Euarthropoda has proven relatively stable, the branching order of clades crownwards relative to radiodonts is much more controversial (see [6]) and heavily depends on the interpretation of the segmental affinity of the head appendages in radiodonts, megacheirans, bivalved euarthropods and fuxianhuiids (figure 1). For instance, radiodont frontal appendages have been regarded as either protocerebral [18,22] or deutocerebral (e.g. [17,21,23]); megacheiran great appendages have also mainly been considered as either protocerebral (e.g. [18]) or deutocerebral (e.g. [21,23,24]); finally, the specialized post-antennal appendages of fuxianhuiids have been regarded as protocerebral, deutocerebral, tritocerebral [25] or even post-tritocerebral [17,26]. Despite these conflicting evolutionary scenarios, it is possible to trace several derived key innovations relative to radiodonts, but the order of their appearance remains highly contentious. Regardless of their precise position in the euarthropod stem-group, bivalved euarthropods, fuxianhuiids, megacheirans and *Kylinxia* feature a fully arthrodized body, a multisegmented head with differentiated deutocerebral appendages and biramous post-oral limbs (e.g. [3,6,9,20,26]). In this context, the post-oral appendage morphology of several Cambrian stem-group euarthropods share structural similarities—such as the presence of endopods with over a dozen podomeres in several representatives—and can inform the autecology of these phylogenetically early forms. In the present contribution, we describe isolated appendages that can be ascribed to previously unknown stem-group euarthropods from the Guanshan Biota (Cambrian: Stage 4) of South China, including a fuxianhuiid, and a new problematic taxon with possible close affinities with either radiodonts or fuxianhuiids.

## 1.1. Geological setting

The Wulongqing Formation of eastern Yunnan hosts the Guanshan Biota, which is known from over 10 localities [27,28]. The Wulongqing Formation comprises mud, silt and sandstones [27,29]. Exceptionally preserved fossils are most commonly found in mudstone units, which probably represent rapid burial events [27,30]. Importantly, sedimentological and trace fossil evidence suggests that the levels hosting exceptionally preserved fossils were probably deposited in a relatively shallower setting compared with all other Cambrian Konservat-Lagerstätten, including the geographically close Chengjiang (Stage 3) and Kaili (Wuliuan) biotas [27,29,31,32]. Euarthropods dominate the biodiversity in the Guanshan Biota [27,33] and are known alongside taxa from numerous other animal groups including annelids, brachiopods, echinoderms, lobopodians and sponges, as well as possible algae [27,28,33]. Notably, despite its geographic proximity to other Konservat-Lagerstätten, the radiodont fauna is endemic, probably a consequence of its shallower and more proximal environment geographically and temporally comparable faunas during Cambrian Stage 4 [34].

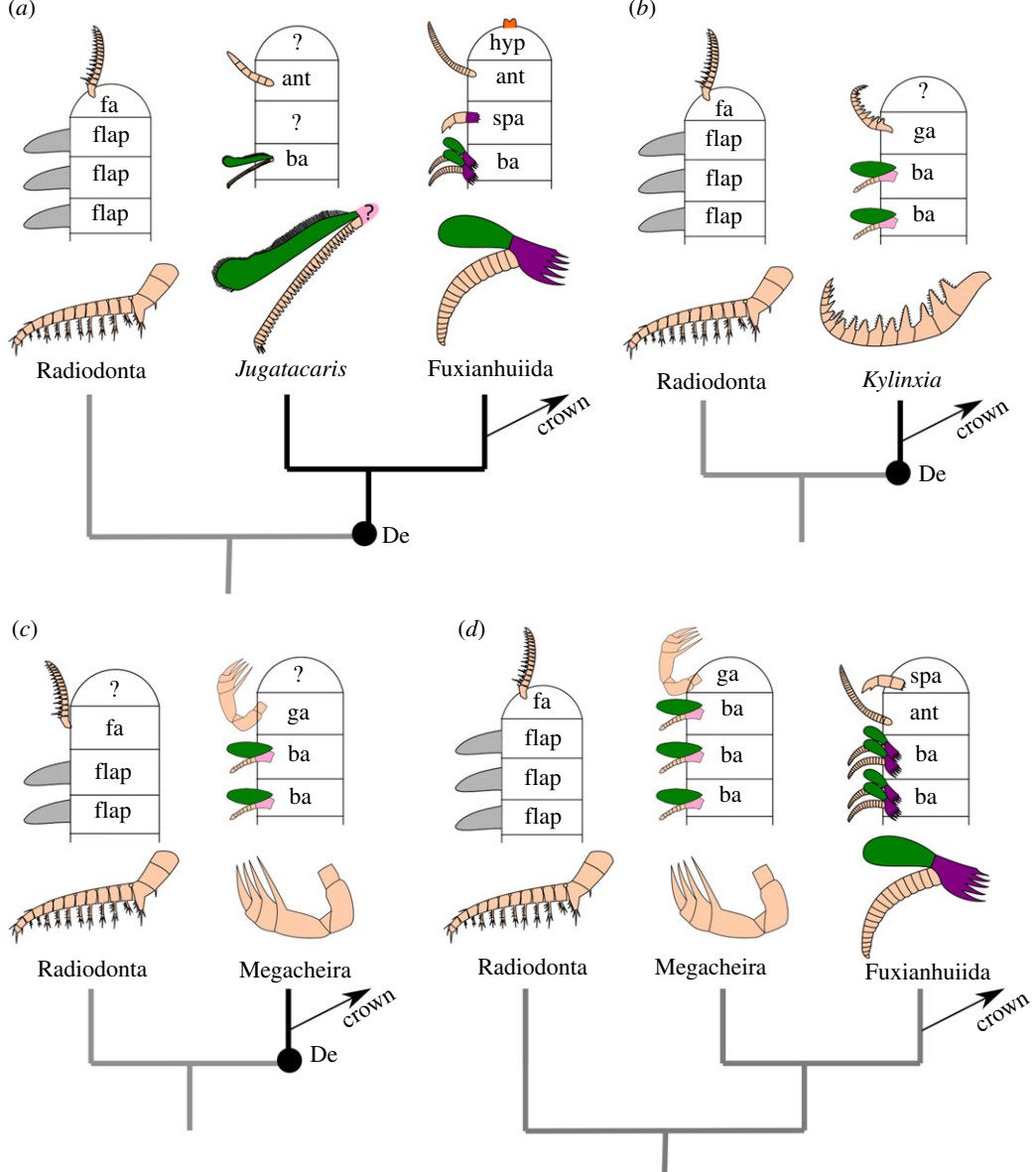

**Figure 1.** Proposed segmental affinities of stem-group euarthropod appendages and resulting hypotheses of stem euarthropod relationships. (*a*) Radiodont frontal appendages protocerebral, megacheiran great appendages deutocerebral, fuxianhuiid specialized post-antennal appendages tritocerebral (e.g. [7]). Deuteropods including paraphyletic bivalved euarthropods such as *Jugatacaris* and fuxianhuiids occupy position immediately crownwards of radiodonts (morphological matrix of e.g. [3,16]). (*b*) Radiodont frontal appendages protocerebral, megacheirans and *Kylinxia* great appendages deutocerebral [9]. *Kylinxia* occupies position immediately crownwards of radiodonts (morphological matrix of [9]). (*c*) Radiodont frontal appendages and megacheiran great appendages both deutocerebral [17]. Megacheirans occupy position immediately crownwards of radiodonts (morphological matrix of e.g. [17]). (*d*) Radiodont frontal appendages, megacheiran great appendages and fuxianhuiid specialized post-antennal appendages all protocerebral [18]. Megacheirans occupy position immediately crownwards of radiodonts (phylogenetic support untested). Black dot marks the base of Deuteropoda in all trees. Radiodont frontal appendage (*Houcaris saron*) redrawn from [19]: figure 1; *Jugatacaris* redrawn from [20]: figure 14; fuxianhuiid appendage (*Alacaris*) redrawn from [16]; *Kylinxia* great appendage redrawn from [9]: figure 3B; megacheiran great appendage (adult *Yohoia tenuis*) redrawn from [21]: figure 10D. Abbreviations: ant, antennae; ba, biramous appendage; De, Deuteropoda; fa, frontal appendage; ga, great appendage; hyp, hypostome/labrum complex; spa, specialized post-antennal appendage.

The material described herein was collected from the Lihuazhuang section of a locality *ca* 2.5 km southeast of Lihua village (figure 2), alongside a diverse radiodont fauna [34], abundant brachiopod valves and trilobite sclerites. The levels of interest belong to the Cambrian Stage 4 (Canglangpuan regional stage; *Palaeolenus* to *Megapaleolenus* trilobite zones) [27].

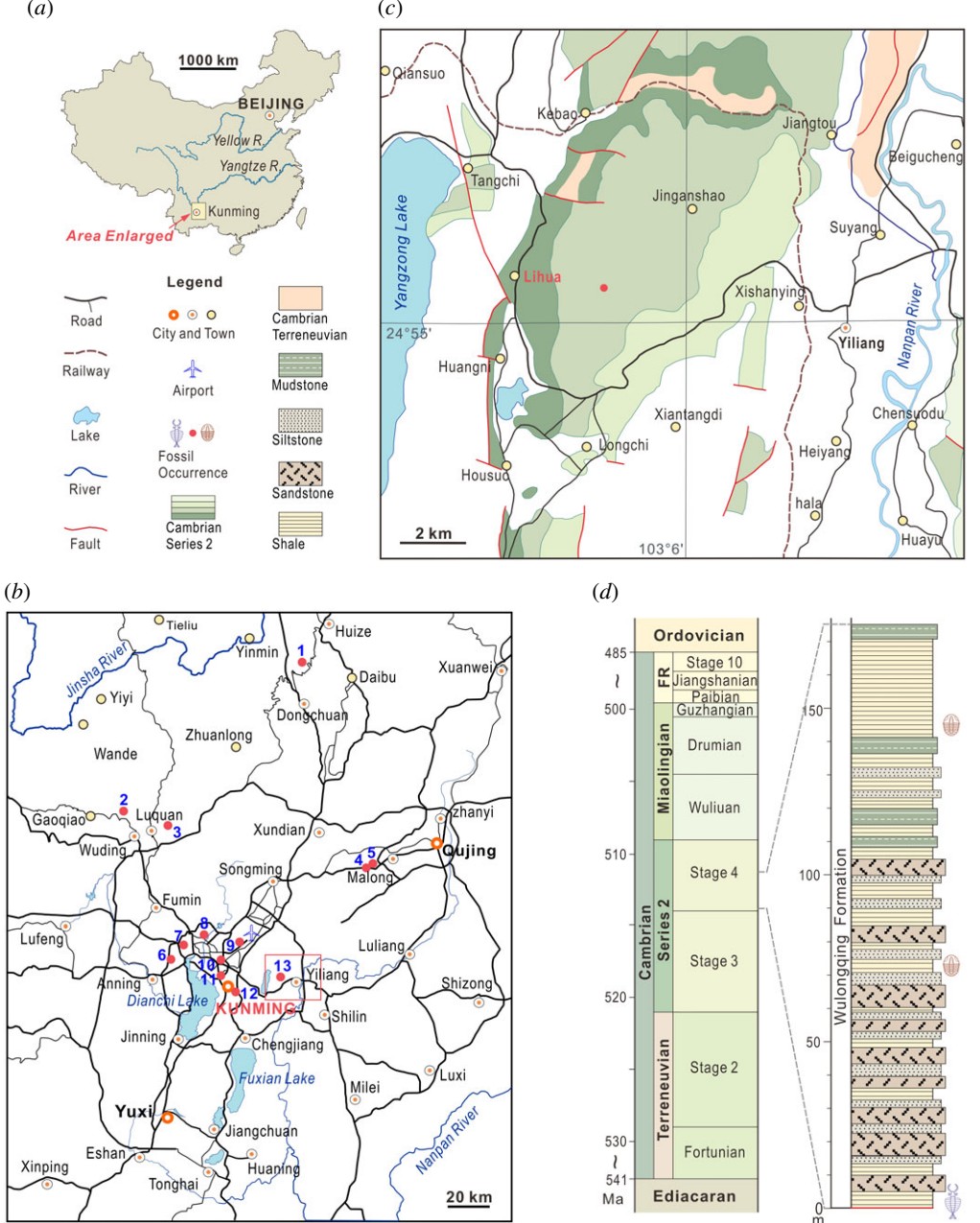

**Figure 2.** Locality maps showing occurrences of known Guanshan biotas in South China, and stratigraphic succession of the Cambrian (Stage 4) Lihuazhuang section in Yiliang, Yunnan. (*a*) Studied area in South China. (*b*) Location of 13 sites where fossil assemblages assigned to the Guanshan Biota have been recovered. (*c*) Enlarged area from (*b*), showing the location of the Lihuazhuang section in Yiliang. (*d*) Stratigraphic column showing the occurrence of fossils reported in this study. Figure reproduced from [34]: figure 1, licensed under CC BY 4.0 (https://creativecommons.org/licenses/by/4.0/).

# 2. Material and methods

All fossil specimens are deposited at the Key Laboratory for Palaeobiology, Yunnan University, Kunming, China (YKLP). A Leica M205-C stereomicroscope with Leica DFC 500 digital camera was used to image specimens. Adobe Photoshop CS 4 and Inkscape 1.0 were used to process photographs, construct line drawings and compile figures. ImageJ2 was used to digitally measure distances from photographs [35].

## 2.1. Terminology

Terminology for the appendages of *Lihuacaris ferox* gen. et sp. nov. follows the orientation commonly used for radiodont appendages (e.g. [36]). We treat the surface bearing outgrowths (endites) as

ventral, with the opposite surface as dorsal. The proximal part of *Lihuacaris* appendages may be a unique structure or homologous to either the shaft of radiodonts or the protopodite of euarthropod biramous limbs. Accordingly, we refrain from using the terms 'shaft', 'distal articulated region' or 'protopodite' for *Lihuacaris*, and instead use 'base' for the morphologically distinct proximalmost element and 'podomeres' for all the others, with the exception of the spinose terminal part that is referred to as the 'distal claw element'.

Fuxianhuiid appendage descriptions follow standard euarthropod terminology [16,25,37,38]. The term 'protopodite' refers to the proximal part of the biramous limb that carries the rami [39]. Measurements of length and height refer to the long axis and short axis of the appendage in lateral view, respectively.

## 2.2. Systematic palaeontology

SUPERPHYLUM Panarthropoda Nielsen [40]

ORDER Fuxianhuiida Bousfield [41]

FAMILY Chengjiangocarididae Hou and Bergström [42]

**Type genus.** *Chengjiangocaris* Hou & Bergström [43]
  **Other genera included.** *Alacaris* Yang, Ortega-Hernández, Legg, Lan, Hou & Zhang [16]

*Alacaris* Yang, Ortega-Hernández, Legg, Lan, Hou & Zhang [16]

*Alacaris*? sp.

**Material.** Two complete isolated endopods (YKLP 12434 and YKLP 12435) and two incomplete endopods (YKLP 12436 and YKLP 12437).

**Locality and horizon.** Lihuazhuang section, locality *ca* 2.5 km southeast of the Lihuazhuang village (figure 2). Cambrian Stage 4, lower part of Wulongqing Formation, *Palaeolenus* biozone [33].

**Description.** Isolated appendages possess an enlarged subtrapezoidal proximal region that is interpreted as the protopodite (figures 3–5), which partly overlies a series of at least 20 subrectangular-shaped endopod podomeres that taper distally (figures 3 and 4). YKLP 12435 shows that the protopodite forms an enlarged gnathobase composed of five robust triangular teeth that face adaxially (figure 4). Two specimens (YKLP 12435, YKLP 12437) reveal that the protopodite also bears two elongate spines that project perpendicular to its ventral margin (2.5 and 5 mm long in YKLP 12435, 2 mm long in YKLP 12437; figures 4 and 5*c*). Two grooves diverge from a common point close to the proximal margin of the protopodite (figures 3–5). One groove (5 mm long in YKLP 12437, 11 mm long in YKLP 12434) follows the proximal two-thirds of the ventral margin of the protopodite before meeting it at the point where the spines project (figures 3 and 5). The second groove (23 mm long in YKLP 12437, 32 mm long in YKLP 12434) runs along with the midline of the protopodite and all visible endopod podomeres. The podomeres remain the same length along the endopod (approx. 1 mm in YKLP 12437, approx. 1.5 mm in YKLP 12434), but become progressively reduced in height distally (from pd3 approx. 6 mm to pd19 approx. 2.5 mm in YKLP 12437, and pd6 approx. 7 mm to pd20 approx. 2.5 mm in YKLP 12434). A single curved subtriangular endite is observed on some podomeres of YKLP 12435 and YKLP 12436 (figures 4 and 5*b*). There is no indication of a distal claw, but instead a rounded termination; it is uncertain if this represents the distal end of the limb or if the latter is simply not preserved in the available material.

Details of the rest of the body are not preserved. YKLP 12435 is associated with various aligned sclerotized plates, most likely trunk tergites based on the chevron-shaped outline (figure 4). YKLP 12436 is associated with the carapace of an unidentified bivalved euarthropod with possible traces of bioturbation (figure 5*a*).

**Remarks.** We identify YKLP 12434, YKLP 12435, YKLP 12436 and YKLP 12437 as isolated fuxianhuiid appendages based on the subtriangular shape of the endopod and presence of more than a dozen tall subrectangular podomeres. The overall subconical shape and elongate outline of these appendages, the presence of a median groove and the enlarged gnathobasic protopodite strongly suggest affinities within the family Chengjiangocarididae [16,25,37] among the order Fuxianhuiida (see also [38]).

*Guangweicaris spinatus* (family Fuxianhuidae) represents the only fuxianhuiid described from the Guanshan Biota to date. This taxon is known from completely articulated specimens that reveal the structure of the trunk appendages in enough detail to warrant direct comparisons with the new material [44,45]. The new endopods described here are inconsistent with those of *Guangweicaris*,

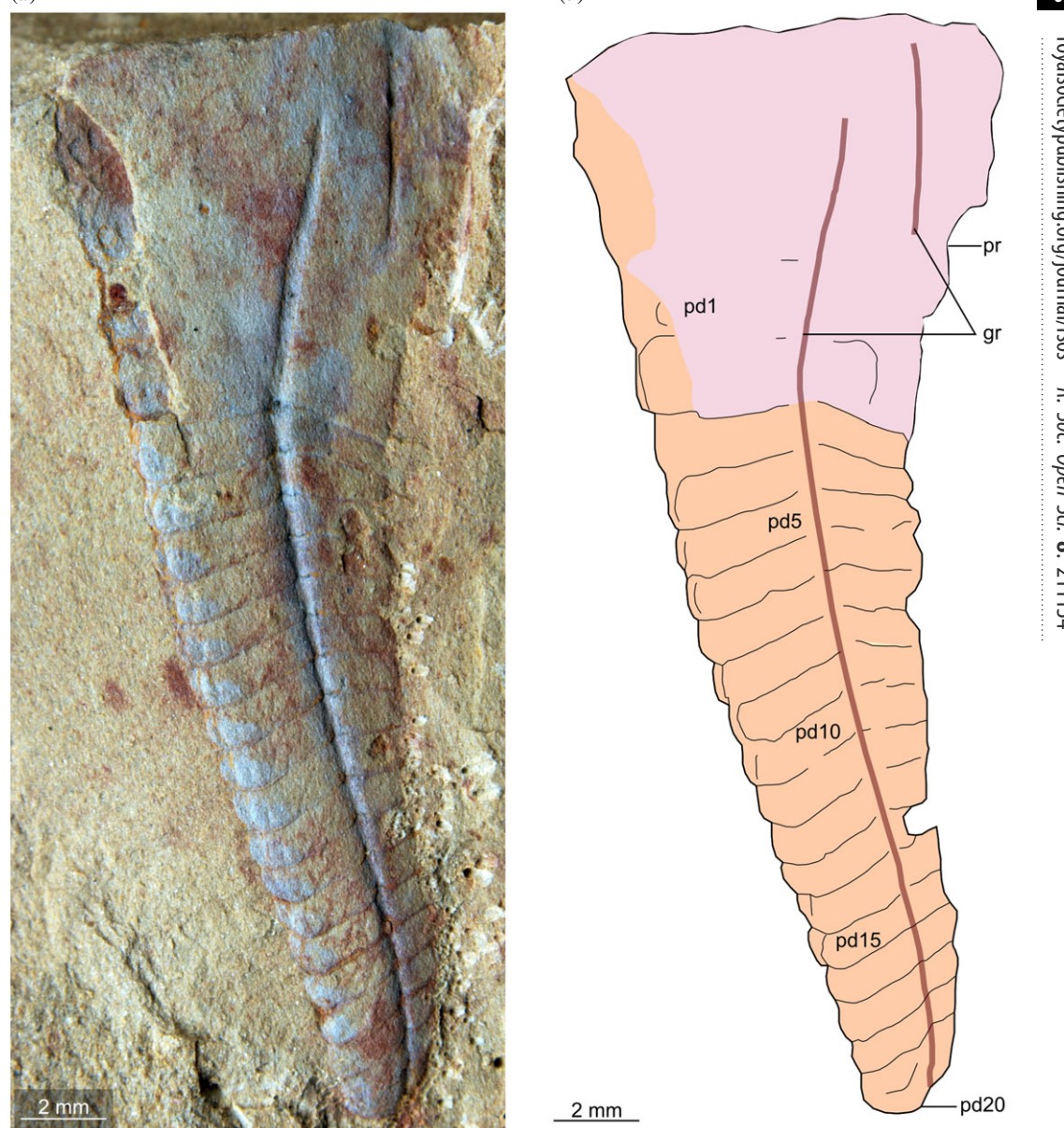

**Figure 3.** *Alacaris*? sp. from the Cambrian (Stage 4) Guanshan Biota, Yunnan, China. YKLP 12434, an isolated endopod with partial protopodite. Abbreviations: gr, linear groove; pd, podomere; pr, protopodite.

including in the presence of up to 20 podomeres (*contra* 11 in *Guangweicaris*), and a well-defined protopodite with gnathobasic edges (absent in *Guangweicaris*). Within Fuxianhuiidae, the Guanshan material shows the closest similarities with the recently described *Xiaocaris luoi* from Chengjiang [38], including the presence of over a dozen endopod podomeres and subtriangular endites. However, *Xiaocaris luoi* clearly lacks a differentiated gnathobasic protopodite and a median groove, and thus the Guanshan material cannot be assigned to this species.

   Among Fuxianhuiida, an enlarged gnathobasic protopodite has only been confirmed in the chengjiangocaridid *Alacaris mirabilis* from the Cambrian (Stage 3) Xiaoshiba biota [16,46]. The spinose margin of the gnathobase in both *Alacaris mirabilis* and the new Guanshan fuxianhuiid appears to bear five medially oriented teeth, although these teeth are more elongate and recurved in the Xiaoshiba material [16]. The tentative assignment to *Alacaris* is supported by the presence of a gnathobasic protopodite, a groove running through the midline of the endopod and a high podomere count. However, the type material of *Alacaris mirabilis* does not show evidence of the paired elongate spines on the protopodite and in the absence of additional information on the exoskeleton in the Guanshan appendages (e.g. hypostome, number of trunk tergites and morphology of the specialized

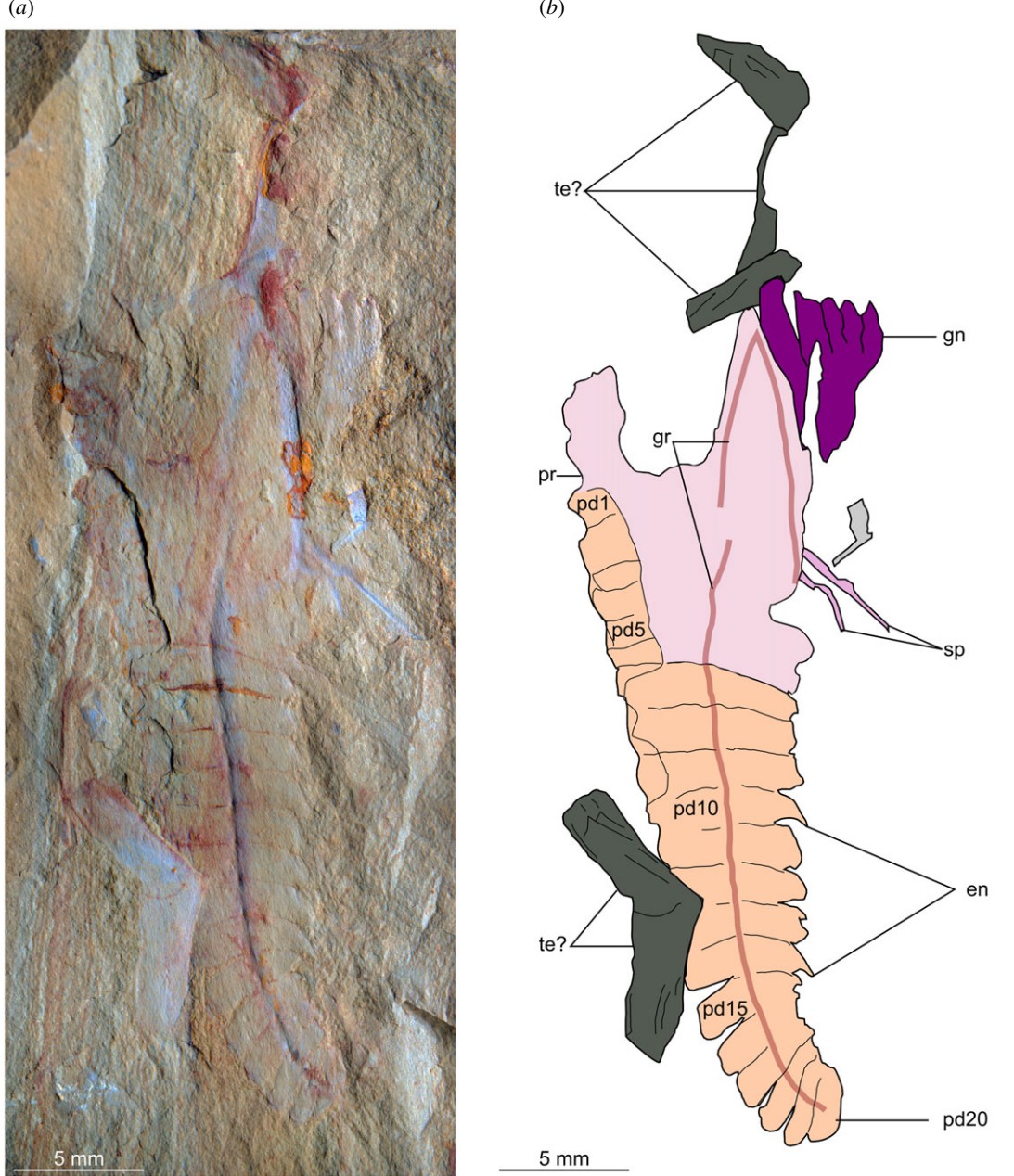

(a)    (b)

**Figure 4.** *Alacaris*? sp. from the Cambrian (Stage 4) Guanshan Biota, Yunnan, China. YKLP 12435, an isolated endopod with gnathobasic protopodite. Abbreviations: en, endite; gn, gnathobase; gr, linear groove; pd, podomere; pr, protopodite; sp, slender spine; te?, possible euarthropod tergite.

post-antennal appendages) cannot be verified. If an assignment to *Alacaris* is confirmed, the higher podomere count, and the presence of a second protopoditic groove and elongate spines in the Guanshan material (not known in the Xiaoshiba material) would warrant the erection of  species. The associated possible tergites are not considered to belong to the same individual, because they are smaller than the endopods. Fuxianhuiid tergites are typically much larger than the associated endopods, as each tergite can cover two or more pairs of biramous appendages (e.g. [46,47]), though it is possible (albeit unlikely based on their triangular shape) that they could be disarticulated reduced anterior trunk tergites that characterize all fuxianhuids. These specimens represent the stratigraphically youngest record of Chengjiangocarididae to date, as well as the only occurrence of this family beyond the Chengjiang (e.g. [42]) and Xiaoshiba biotas [16,25], both of which are geographically located in the vicinity of Kunming, thus extending the known temporal range of the family from the Cambrian Stage 3 to Stage 4, and geographically towards eastern Yunnan.

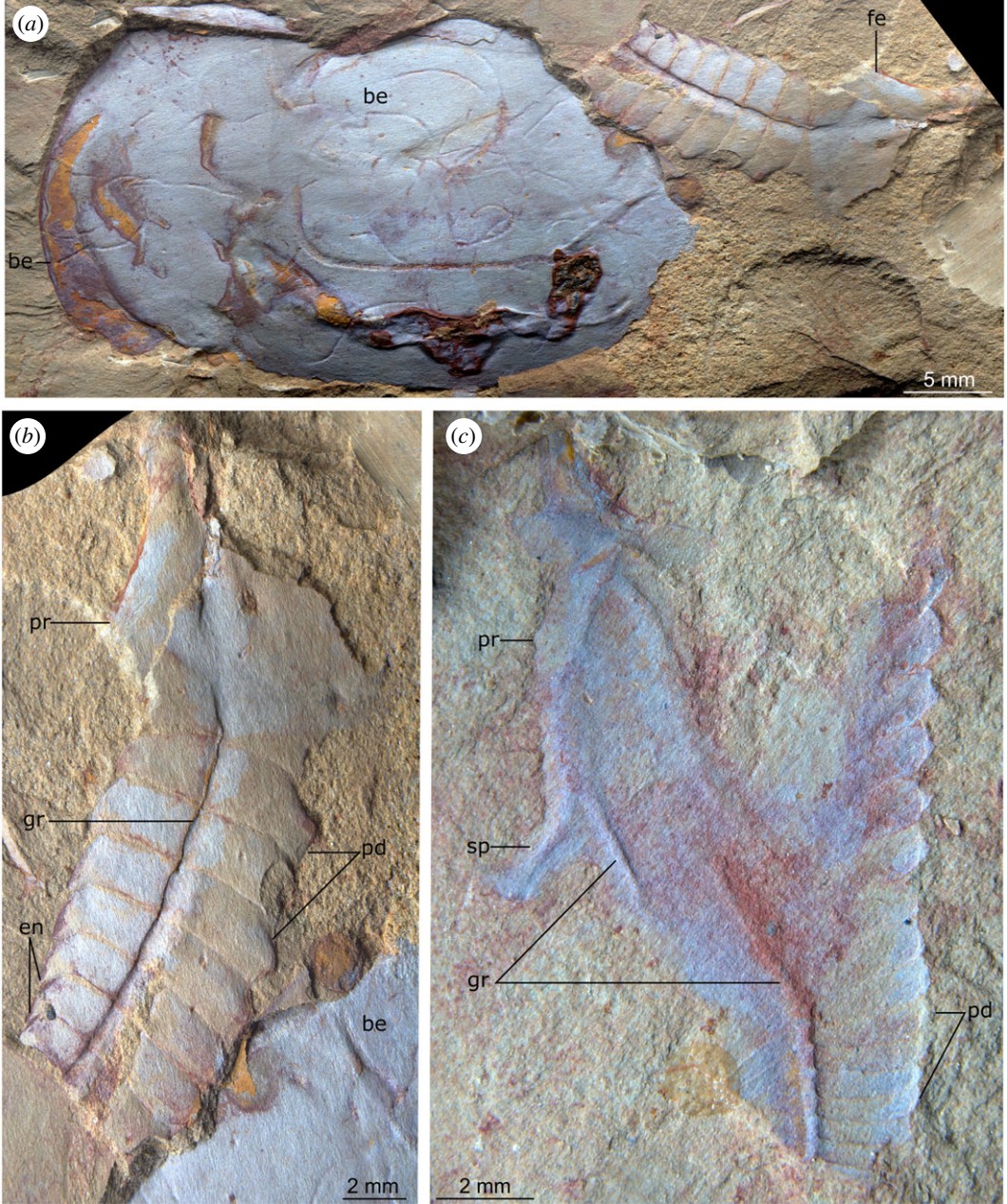

**Figure 5.** *Alacaris*? sp. from the Cambrian (Stage 4) Guanshan Biota, Yunnan, China. (*a,b*) YKLP 12436, an isolated endopod with partial protopodite adjacent to a 'bivalved euarthropod'. (*c*) YKLP 12437, an isolated endopod with partial protopodite. Abbreviations: be, 'bivalved euarthropod' carapace; en, endite; gr, linear groove; pd, podomere; pr, protopodite; sp, slender spine.

Incertae sedis

GENUS *Lihuacaris* gen. nov.

**Type species.** *Lihuacaris ferox* gen. et sp. nov.

**Diagnosis.** Arthropodized elongate appendage composed of a rectangular base, proximal relative to 11 tall rectangular podomeres and a long subtriangular distal element; podomeres increase in length distally, alternate with triangular articulating membranes and bear small triangular endites (one pair per podomere) that insert at the midpoint of ventral margin; distal element bears 8–10 robust curved ventral spines that increase in size towards the distal end of the appendage.

**Etymology.** Concatenation of the first part of the name of the section where the fossils were found (Lihuazhuang), and the Latin 'caris' (or Greek 'καρις'), meaning 'crab' or 'shrimp', a suffix commonly used for euarthropods.

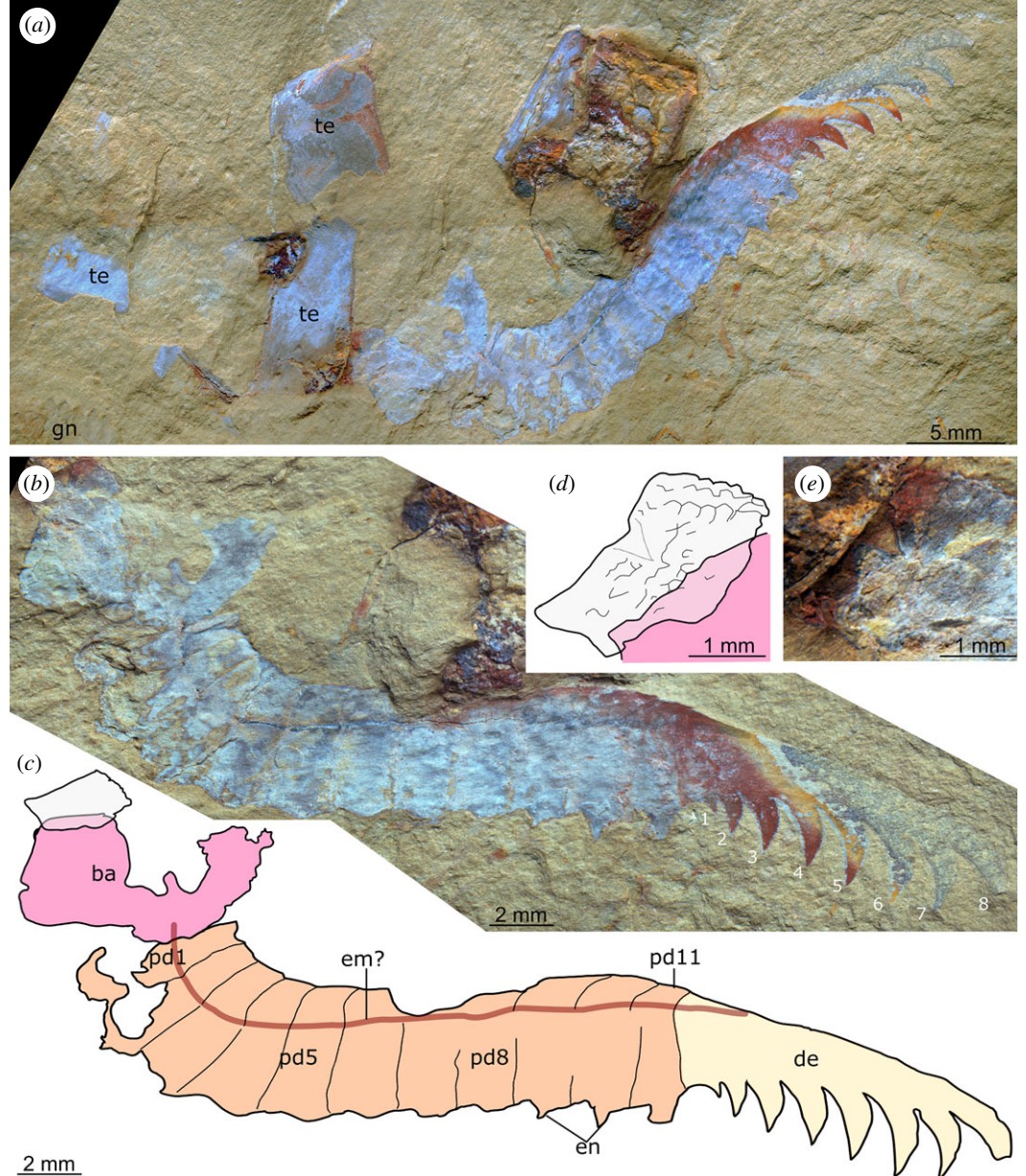

**Figure 6.** *Lihuacaris ferox* nov. gen et sp. from the Cambrian (Stage 4) Guanshan Biota, Yunnan, China. YKLP 12438, holotype, an isolated appendage preserved laterally compressed. (*a*) Overview of slab showing appendage and associated with euarthropod tergites. (*b,c*) Details of appendage. (*d,e*) Details of rectangular structure which is overlain by the large base of the appendage. Abbreviations: ba, large rectangular base; de, distal element; em?, possible extensor muscle; en, endite; gn, gnathobase; pd, podomere; te, tergite; 1–8 indicate spines on distal element, numbered from proximal to distal.

**Type material.** *Holotype:* YKLP 12438 (figure 6), a complete appendage preserved as a lateral compression; *Paratype:* YKLP 12439 (figure 7*a*), an incomplete appendage missing the base and proximalmost six podomeres, preserved as a lateral compression.

**Additional material.** Four partial specimens YKLP 12440–12443 (figure 7*b–e*).

**Type locality and horizon:** Lihuazhuang section, locality *ca* 2.5 km southeast of the Lihuazhuang village (figure 2). Lower part of Wulongqing Formation, Cambrian Stage 4, *Palaeolenus* biozone [33].

*Lihuacaris ferox* sp. nov.

**Diagnosis.** As for genus, by monotypy.

**Etymology.** From *ferox* (Latin = ferocious) in reference to the inferred predatory habits of this animal.

**Description.** The new taxon is known only from isolated, mostly incomplete appendages. The holotype is preserved associated with a number of rectangular sclerotized tergites of uncertain

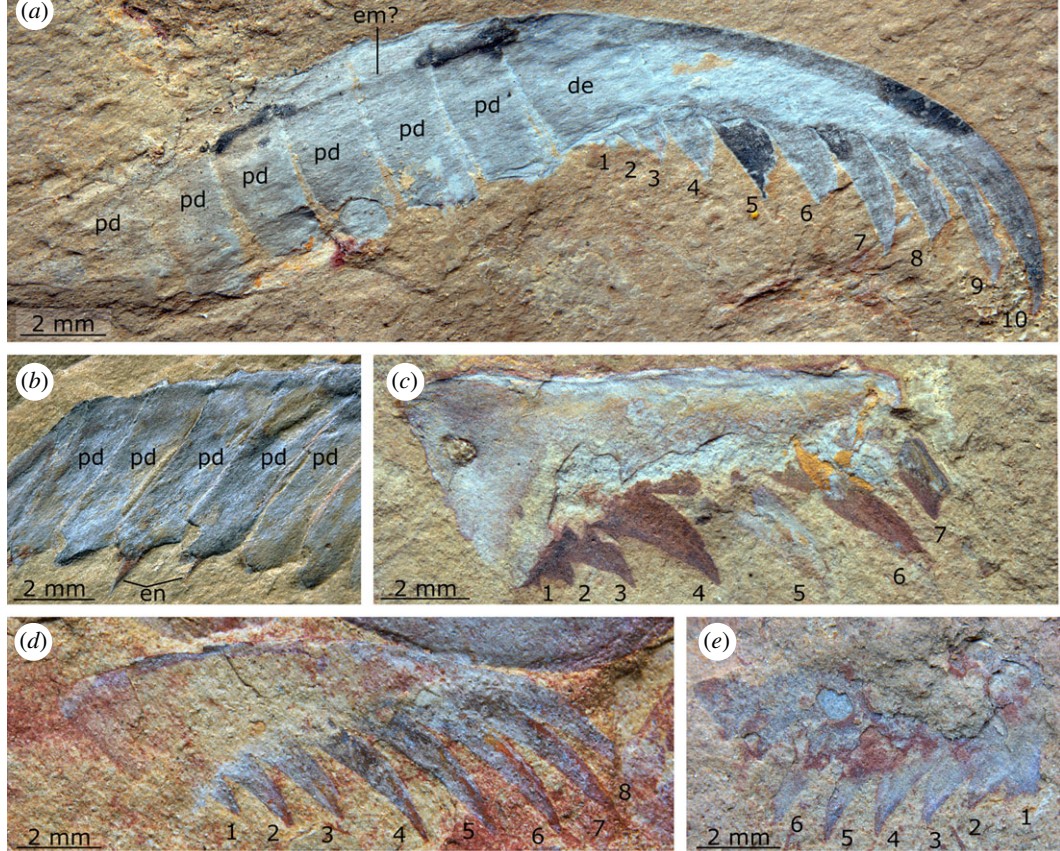

**Figure 7.** *Lihuacaris ferox* nov. gen et sp. from the Cambrian (Stage 4) Guanshan Biota, Yunnan, China. (*a*) YKLP 12439, paratype, a partial isolated appendage showing distal podomeres and distal element. (*b*) YKLP 12440, a partial isolated appendage showing podomeres and endites. (*c*) YKLP 12441, a partial distal element. (*d*) YKLP 12442, a complete distal element E. YKLP 12443, a partial distal element. Abbreviations: de, distal element; em?, possible extensor muscle; en, endite; pd, podomere; 1–10 indicate spines on distal element, numbered from proximal to distal.

affinities and an isolated gnathobase (figure 6*a*). A small rectangular structure with a polygonal texture is overlain by the proximal part of the appendage (figure 6*d,e*). The size and position (underneath) of this structure relative to the rectangular base of the appendage suggest that it does not belong to the appendage proper. An additional partial appendage is associated with a putative bivalved euarthropod carapace (figure 7*d*).

The holotype (YKLP 12438) of *Lihuacaris* is an isolated appendage that measures *ca* 28 mm along its dorsal margin (from the proximal part of pd1 to the tip of the distal element). The appendage consists of a rectangular base, proximal to 11 podomeres (pd1–11) and a large spinose distal element (figure 6). The triangular articulating membranes that separate the podomeres are wider ventrally and taper to a point dorsally. The remaining specimens represent isolated distal elements, isolated podomeres or a combination of both (figure 7). Distal to the base, the appendage curves dorsally at an angle of *ca* 50 degrees from pd1–5; its height decreases by 20%, while podomere length increases by 30%, resulting in the podomere height/width ratio to decrease from four to two. From pd6 to 10, the appendage displays a straight dorsal margin; podomeres are roughly similar in their proportions to pd5, except for a slight increase in length. Pd11 is distinctly shorter than pd10 dorsally, which gives it a trapezoidal outline. Short, triangular endites project from the midpoint of the ventral surface of at least pd7–9, whereas pd10 is associated with a similarly located, rectangular remains that looks like the proximal part of a larger endite (figure 6*b,d*). Evidence that these endites are paired comes from an additional specimen (YKLP 12440) that preserves only the podomeres (figure 7*b*). The distal element is approximately the same length as the six distalmost podomeres combined. This structure is best preserved in the paratype (YKLP 12439; figure 7*a*), which shows that it has a smooth dorsal margin that curves ventrally. The ventral margin is similarly curved and bears 8–10 spines, which are slightly curved proximally and increase in length toward the appendage tip (figures 6 and 7).

Proximal spines in the paratype appear broken close to the base (figure 7a), while brittle fractures run through isolated distal elements (figure 7c,d). In the holotype (figure 6), the distal element bears only eight ventral spines, but the latter structures are more widely spaced. Two small, incomplete, isolated distal elements (figure 7c,e) bear shorter hooked spines relative to the size of the distal element (even considering its incomplete nature) compared with the larger distal elements (figures 6 and 7a,d). A dark-coloured trace runs parallel to the long axis of the appendage from pd1 to the distal element in the holotype and, to a lesser extent, the incomplete paratype (figures 6 and 7a). It starts off at the mid-height of pd1 (figure 6) and progressively converges toward the dorsal margin that it borders in the distal element (figure 7a). The location of this structure in the dorsal half of the appendage and the fact that it connects the base to the dorsal margin of the distal element is suggestive of an extensor muscle.

## 3. Discussion

### 3.1. Affinities of Lihuacaris ferox

The morphology and preservation of *Lihuacaris ferox* meet all the criteria for arthropodized appendages [48], namely the presence of sclerotization, segmentation and functional articulations. Sclerotization is attested by the limited evidence for soft deformation and the presence of brittle fractures. The appendages are clearly serially segmented and articulated with pivot-like joints. Arthrodial membranes are expressed as triangular material that is more faintly preserved than the rectangular blocks of heavily sclerotized cuticle. Each of the triangular articulating membranes narrows to a pivot point dorsally, consistent with the morphology of flexible euarthropod appendages (e.g. [39]).

Radiodonts are the earliest diverging stem-group euarthropods with arthropodized appendages [5,49,50], and they were among the largest animals in Cambrian oceans, ranging from millimetres to around a metre in length (e.g. [51,52]). Many representatives of this group, including those from the Guanshan Biota (e.g. [34]), possess frontal appendages that could reach a length comparable to *Lihuacaris* of up to several centimetres. *Lihuacaris* shares discrete morphological characters with radiodonts, including the triangular shape of arthrodial membranes, and the presence of a single endite pair attaching medially within the podomeres. Amplectobeluid radiodonts, such as *Amplectobelua symbrachiata*, possess appendages of a similarly elongate shape and a comparable number of tall rectangular podomeres (figure 8a) [56]. Endites similar in size (relative to the size of the appendage) and shape (triangular) to those of *Lihuacaris* can also be found in the Burgess Shale amplectobeluid *Amplectobelua stephenensis* and representatives of *Caryosyntrips*, a radiodont genus of three species recovered from deposits in Spain and North America [49,57–59]. Lastly, the putative extensor muscle of *Lihuacaris* passes through the same part of the appendage (close to the dorsal margin) and is similar in size and curvature to the 'diffuse band' described from *Anomalocaris canadensis* frontal appendages [50]. However, other aspects of the organization of *Lihuacaris* appendages preclude a radiodont affinity.

*Lihuacaris* appendages can be broadly separated into three regions: a large base, an intermediate region of numerous endite-bearing podomeres and an elongate distal element (figure 8b). The first two regions can be broadly compared with the shaft and distal articulated region of radiodonts, which are distinguished from one another by the presence in the latter region of well-developed articulating membranes. Indeed, various radiodonts display a monosegmented shaft comparable to *Lihuacaris ferox* (e.g. *Anomalocaris canadensis*; 'Anomalocaris' briggsi; *Peytoia nathorsti*; *Caryosyntrips serratus* [49,50,60,61]), though multisegmented shafts are also known (e.g. *Houcaris saron*; *Laminacaris chimera*; *Ramskoeldia platyacantha* [36,62,63]; but see [64] for an alternative view). Just as in *L. ferox*, radiodont shaft podomeres can also be of different shape and size to those in the distal articulated region. For example, the shaft podomere of 'Anomalocaris' briggsi is larger than the other podomeres of the appendage [61]. However, a long and broad rigid distal element is as-yet totally unknown in radiodonts, as is the presence of more than a single endite/pair of endites per appendicular element (i.e. podomere in radiodonts). The distal element in *Lihuacaris* superficially resembles complete appendages of *Caryosyntrips* species, which are similarly elongate, triangular (but not distally recurved) and interpreted as functionally non-articulating [49,57]. However, although usually incomplete, podomere boundaries can be distinguished in some *Caryosyntrips* appendages, which indicate their likely origin from a multisegmented articulated appendage. *Caryosyntrips* specimens also display a bell-shaped proximal margin that suggests a broad, highly flexible contact with a much

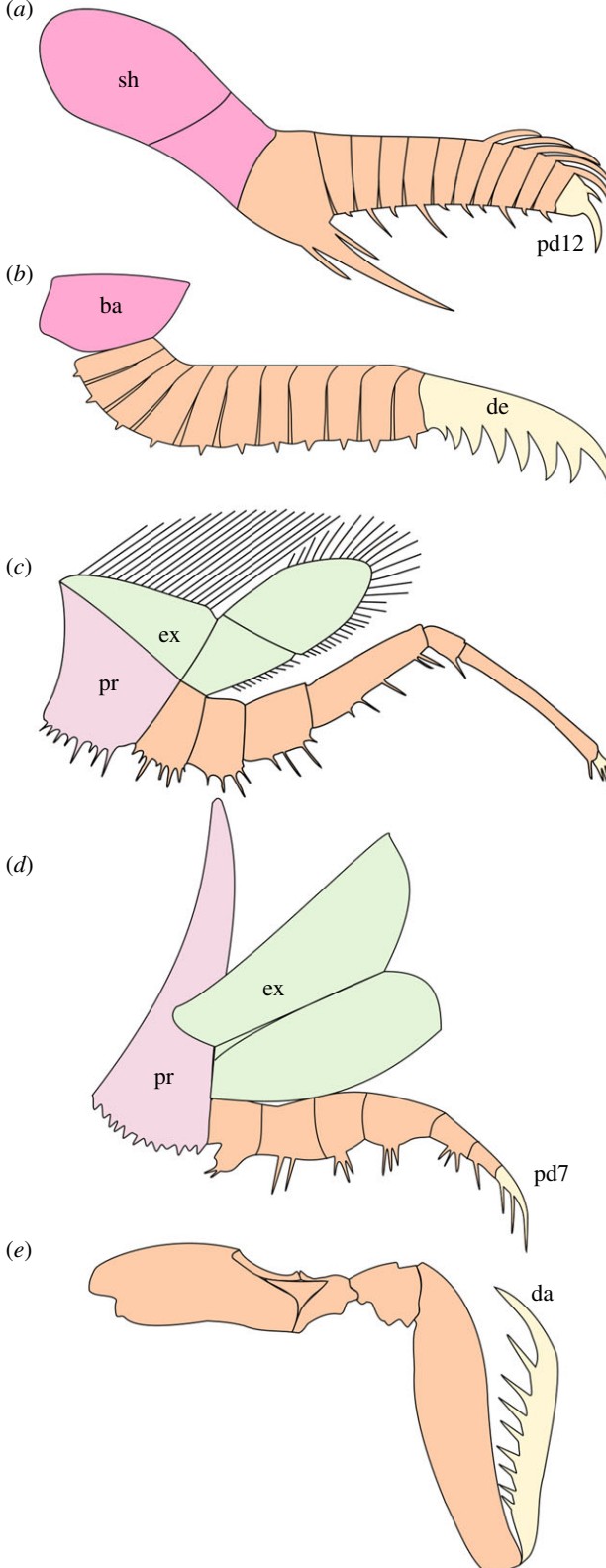

**Figure 8.** Appendages of total-group euarthropods. Colours indicate parts of appendage compared in the text. (*a*) Protocerebral frontal appendage of amplectobeluid radiodont *Amplectobelua symbrachiata*. (*b*) Appendage of uncertain segmental affinity of *Lihuacaris ferox* gen. et sp. nov. (*c*) Biramous second trunk limb of artiopod *Emeraldella brocki*. (*d*) Post-antennal appendage of artiopod *Sidneyia inexpectans*. (*e*) Raptorial appendage of spearing-type mantis shrimp *Lysiosquilina maculata,* in lateral view. Outline (*a*) adapted from [19]: figure 6 g. Outline (*c*) redrawn from [53]: figure 9C. Outline (*d*) adapted from [54]: figure 2. Outline (*e*) adapted from [55]: figure 4B. Abbreviations: ba, base; da, dactylus; de, distal element; ex, exopod; pd, podomere; pr, protopodite; sh, shaft.

larger body part (most likely the head), as illustrated by a unique association of two of those appendages forming a pincer-like apparatus [49]—and so would not be expected to attach to an appendage with multiple articulations as displayed by *Lihuacaris*. The distal element of *Lihuacaris* is comparable to the terminal claw of some radiodonts. Some amplectobeluids bear multiple curved spines on the terminal podomere (figure 8*a*) [49], although these are considerably shorter and less numerous than those observed in *Lihuacaris*. This could suggest that the distal element of *Lihuacaris* represents a hypertrophied radiodont terminal claw, given the lack of evidence for segmentation or fusion from multiple podomeres (as in *Caryosyntrips*); however, we refrain from further speculation given the premature understanding of the morphology and affinities of the new taxon. Finally, the dorsal flexure displayed by the proximal podomeres of the holotype of *Liuhacaris ferox* superficially recalls the dorsal kink separating shaft and distal articulated region in some radiodonts. This characteristic feature of amplectobeluids and a few anomalocaridids [19,22,51,56,62,65] can also be observed in *Kylinxia* [9]. The dorsal kink is an abrupt deflection of the long axis of the appendage at the transition from shaft to distal articulating region, very different to the progressive dorsal flexure involving multiple podomeres as seen in *Lihuacaris ferox*. Regarding its general posture, the appendage of the new taxon is more reminiscent of the endopod of a post-antennal appendage of some deuteropods, especially chengjiangocaridid fuxianhuiids (e.g. [16,25,37,42]), as well as some Cambrian members of the class Artiopoda (*sensu* [53]).

Several of the characters observed in *Lihuacaris* have parallels in the well-developed endopods of the biramous appendages of fuxianhuiids, particularly those of members of Chengjiangocarididae. Characters that support a possible chengjiangocaridid affinity include a similar endopod length, multipodomerous condition (*ca* 12 podomeres) and comparable rectangular podomeres with triangular arthrodial membranes between them (e.g. [16,37,42]). Furthermore, the presence of median triangular endites in *Lihuacaris* resembles those of the recently described fuxianhuiid *Xiaocaris luoi* from Chengjiang [38], although it should be noted that the latter species is a member of Fuxianhuiidae, rather than Chengjiangocarididae. The elongate structure interpreted as an extensor muscle in both *Lihuacaris* specimens (figures 6 and 7) is also comparable to the linear groove present in chengjiangocaridid endopods, including those of *Alacaris*? sp. described here (figure 5). However, a possible distinction is that this structure seems to run more dorsally in *Lihuacaris* compared with those of chengjiangocaridids [16,25,37,42]. It is also notable that there are multiple isolated exoskeletal fragments that resemble fuxianhuiid body parts in the vicinity of YKLP 12438. For example, the specimen is associated with remains similar to the trunk tergites of the Guanshan species *Guangweicaris spinatus* [44,45], and an isolated gnathobasic protopodite (figure 6*a*) akin to that of *Alacaris*? sp. (Figure 4) is also found close by. However, in both cases, it is not possible to entirely rule out the possibility that this represents a chance association given the disarticulated nature of the material, and the fact that both *Guangweicaris* [44,45] and *Alacaris* [16] are sufficiently well known from Guanshan and Xiaoshiba material, respectively, to confidently rule out the possibility that *Lihuacaris* might belong to either of those taxa. Despite these similarities, the prominent distal claw of *Lihuacaris* has no parallel with the appendage organization of any fuxianhuiid described to date [16,25,37,42–46].

Artiopods typically exhibit a broad and flat dorsal exoskeleton, with well-developed pleurae ('tergopleurae', 'paratergal folds') that cover a series of laterally splayed biramous appendages. These appendages insert on the ventral surface of the axial region of the body, and therefore, the change of orientation of their endopods from ventro-latero to lateral (or even dorso-lateral) may be associated with a marked dorsal flexure (e.g. [53,66]), somewhat similar to that displayed by the holotype of *Lihuacaris ferox*. However, this dorsal flexure typically involves one or two podomeres in artiopods (figure 8*c*,*d*), not five tall and roughly rectangular ones as in *Lihuacaris*. A comparison to one of these artiopods, *Sidneyia inexpectans*, is also pertinent as this vicissicaudate artiopod has appendages with spinose distal elements (figure 8*d*) [66]. The terminal (seventh) podomere of the endopod of each post-antennal appendage is an elongate, recurved claw-like element, which bears three ventral spines increasing in size distally. However, in addition to bearing fewer spines on this distal element, the endopod of *Sidneyia inexpectans* differs from that of *Lihuacaris* in several important ways: the presence of only seven podomeres, a significant change of podomere shape from tall rectangular to elongate rectangular distally, the split into two morphologically and functionally distinct sets of podomeres (four, tall rectangular to sub-square, proximal ones bearing bunches of slender ventral spines, and three, elongate rectangular, distal ones equipped with stout ventral spines) and the presence of a gnathobase [54,66–69]. The appendage of *Lihuacaris* also differs from artiopod endopods by the presence of a greater number of articulated elements, 13 in total (base and distal claw included)

against eight in artiopods (protopodite included). This observation stands for most Cambrian deuteropods, which typically possess endopods with fewer than 10 podomeres. Fuxianhuiids represent notable exceptions to this, for their multipodomerous limbs may include up to 20 podomeres ([16,20,25,35]; this study). Lastly, the overwhelming majority of post-antennal appendages of Cambrian artiopods, and more generally deuteropods, are biramous, while the appendage of *Lihuacaris* shows no indication of the presence of a second ramus.

*Lihuacaris* also shows similarities with two additional Cambrian euarthropods with enigmatic affinities, namely *Kiisortoqia soperi* from the Stage 3 Sirius Passet [70], and the recently described *Bushizheia yangi* from the Stage 3 Chengjiang [71]. Both taxa have a broad exoskeleton with a cephalon, freely articulating trunk tergites, and in the case of *Bushizheia*, a prominent pygidium, all of which indicate deuteropod affinities. However, they also feature a pair of robust first appendages that evoke similarities to *Lihuacaris* and the frontal appendages of Anomalocarididae, including the presence of more than 10 homonomous podomeres that taper distally and robust spinose endites. *Lihuacaris* differs from these taxa in some key details, such as the presence of a subrectangular base, broad and elongate distal element (both characters absent in *Kiisortoqia* and *Bushizheia*), and the insertion of the spiniform endites at the podomere mid-length (distal to podomere mid-length in *Kiisortoqia* and *Bushizheia*).

In summary, the *Lihuacaris* appendage shows some similarities with Cambrian representatives of both Radiodonta and Deuteropoda, but these features are combined in a unique way that is not readily represented in any known Cambrian total-group euarthropod. Additionally, the distal element of the *Lihuacaris* appendage is unique, which together with the overall morphology of these appendages precludes a precise systematic assignment within total-group Euarthropoda, though a position close to Radiodonta or within early branching members of Deuteropoda is most likely.

## 3.2. Position of *Lihuacaris ferox* in the euarthropod stem-lineage

The presence of a differentiated base in a limb with several morphological similarities to radiodont frontal appendages and fuxianhuiid trunk endopods could suggest affinities within both lower stem-group Euarthropoda and Deuteropoda. However, as it is not clear whether the appendages of *Lihuacaris* have the same segmental affinity as radiodont frontal appendages, a wider range of possibilities should be considered (figure 9). Below we discuss the possible positions of *Lihuacaris* in relation to the different tree arrangements proposed for the euarthropod stem and their phylogenetic implications.

**Lihuacaris** *appendages as protocerebral* (figure 9*a*): this scenario would imply that the *Lihuacaris* appendages are serially homologous with the protocerebral frontal appendages of radiodonts [7,22] and suggest that *Lihuacaris* would either be the earliest diverging stem-group euarthropod with arthropodized appendages, or occupy a crownward position relative to radiodonts (figure 9*b,c*). Positioned at the anterior margin of the head, the base of *Lihuacaris* appendages would have a similar function to the proximal shaft of radiodont appendages, and the remaining podomeres would most likely serve a comparable grasping or raptorial function for feeding to that hypothesized for taxa such as Anomalocarididae and Amplectobeluidae [49–51].

If *Lihuacaris* occupies a more stemward position it could potentially inform the ancestral morphology of Radiodonta, particularly before the evolution of the well-developed armature of specialized ventral endites observed in all representatives of this clade. This position would also imply that the body of *Lihuacaris* shared fundamental similarities with lower stem-group euarthropods, including a limited or absent body sclerotization and presence of lateral body flaps. Alternatively, a crownward placement of *Lihuacaris* relative to radiodonts would cause considerable uncertainty regarding the trunk morphology of this taxon, as the character polarity of dorsal arthrodization and biramous trunk appendages cannot be resolved based on the frontal appendages alone, even though these changes are expected to occur in the node above Radiodonta.

**Lihuacaris** *appendages as deutocerebral* (figure 9*d*): the similarities between the appendages of *Lihuacaris* and some deutocerebral appendages, such as those borne by *Kylinxia* or some megacheirans [9,21,23,24,72] raise the possibility that *Lihuacaris* appendages originate from the deutocerebral segment. *Lihuacaris* appendages are also similar to the first appendages of the enigmatic euarthropods *Kiisortoqia* [70] and *Bushizheia* [71], both of which sit within Deuteropoda based on the presence of a multisegmented head, fully arthrodized trunk and biramous appendages. These comparisons would place *Lihuacaris* crownwards of radiodonts, possibly as an early branching deuteropod similarly to *Kylinxia* [9], while also implying the likely presence of multisegmented head, fully arthrodized trunk

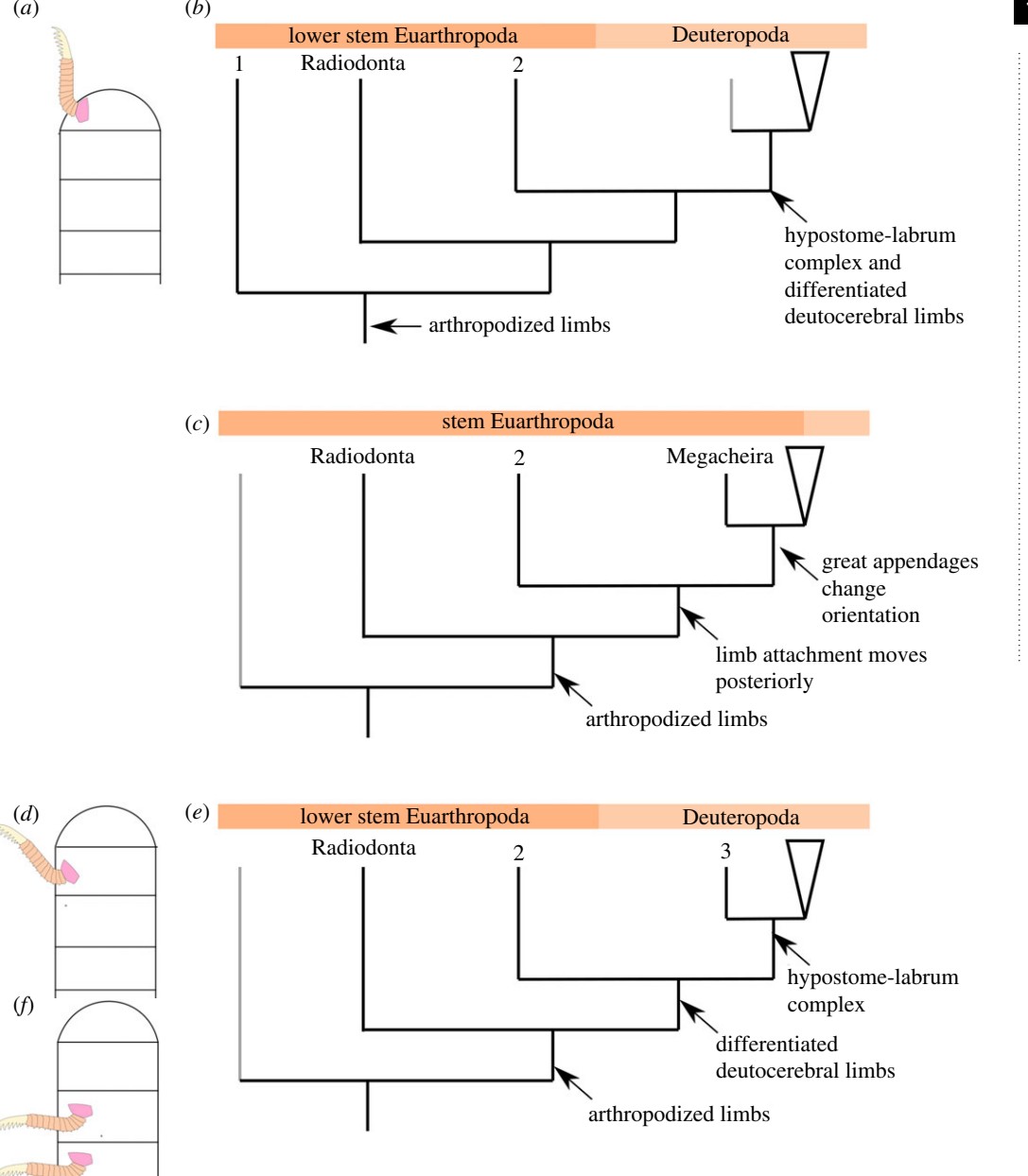

**Figure 9.** Hypothetical phylogenetic and segmental affinities of *Lihuacaris ferox* appendages. (*a*) Protocerebral affinity for *Lihuacaris* appendages. (*b*) Hypothetical relationship of *Lihuacaris* to other stem-group euarthropods, assuming a protocerebral affinity for the appendages, and considering megacheirans and *Kylinxia* as deuteropods (e.g. [6,9]). Either (1) as the earliest diverging stem euarthropod with an arthropodized limb, or (2) between radiodonts and deuteropods. (*c*) Hypothetical relationship of *Lihuacaris* between radiodonts and megacheirans, assuming a protocerebral affinity for the appendages and considering megacheirans and *Kylinxia* to also possess protocerebral appendages [18]. Under this scenario, the base of *Lihuacaris* appendages would have allowed a more posterior attachment to the head than radiodonts. The orientation of the endites suggests a position stemwards of megacheirans and *Kylinxia*. (*d*) Deutocerebral affinity for *Lihuacaris* appendages. (*e*) Hypothetical segmental affinity of *Lihuacaris* appendages assuming either a deutocerebral or post-deutocerebral affinity for the appendages, assuming megacheirans and *Kylinxia* as deuteropods (e.g. [6,9]). Either (2) between radiodonts and deuteropods, or (3) as an early diverging deuteropod. Note: If radiodonts appendages considered deutocerebral (e.g [17]), *Lihuacaris* could also occupy a position stemwards of radiodonts. (*f*) Post-deutocerebral affinity for *Lihuacaris* appendages. Grey bars indicate positions unavailable for *Lihuacaris ferox* for a given hypothetical segmental affinity for the appendages.

and biramous appendages. As demonstrated by the morphology of *Kiisortoqia* and *Bushizheia*, it could also be possible that *Lihuacaris* appendages represent a highly derived raptorial-like morphology that evolved from a modified set of ancestral antenniform deutocerebral limbs.

**Lihuacaris** *appendages as post-deutocerebral* (figure 9*f*): A post-deutocerebral segmental affinity for *Lihuacaris* appendages would also imply a position crownwards of radiodonts (figure 9*e*). In this segmental scenario, the base of *Lihuacaris* appendages could serve a similar function to the protopodite of the biramous limbs, attaching the appendage to the ventral margin of the body. The dorsal curvature of the proximal five podomeres would allow the appendage to change orientation, just as one or two trapezoidal podomeres do in artiopods (discussed above).

If the *Lihuacaris* appendages are considered to be the post-antennal appendages of a deuteropod, it would imply a likely position as an early diverging member of the clade whether considering the tree topology and associated assumptions of Aria *et al.* [17], Ortega-Hernández [6] or Zeng *et al.* [9]. Fuxianhuiids and early diverging euarthropods with bivalved carapaces (e.g. *Nereocaris* and *Jugatacaris*) display multipodomerous biramous appendages with homonomous podomeres [16,20,73], while the post-antennal appendages of megacheirans and *Kylinxia* are also biramous but have fewer podomeres (*ca* seven) in the endopod (e.g. [9,21]). The similarity in dorsal curvature between *Lihuacaris* appendages and artiopods does not imply close affinities, as they would be convergent adaptations to a flattened body plan.

More complete *Lihuacaris* specimens will provide evidence for the segmental affinity of the appendages described herein. This information is required to accurately place this taxon within the euarthropod total-group, which in turn will provide additional information on the polarity of other characters in the euarthropod stem-lineage, such as the presence or absence of dorsal arthrodization (absent in radiodonts, present in the group immediately crownwards), regardless of the favoured tree topology. For now, the isolated *Lihuacaris ferox* appendages offer a hint of the potential of future material from the Guanshan Biota for resolving larger-scale outstanding problems concerning the evolution within total-group Euarthropoda.

## 3.3. Morpho-functional interpretations and feeding ecologies

The appendicular anatomy of *Lihuacaris* shows similarities with those of deuteropods with predatory habits. The distal element of the new taxon resembles the distal region of a post-antennal endopod of *Sidneyia inexpectans* (figure 8*d*). This taxon fed on both hard-shelled and soft prey animals, as revealed by gut content analysis [66,67] and limb biomechanics [54,69], although these adaptations are mostly constrained to the proximal region of the appendages. The distal element of *Lihuacaris* also bears a functional resemblance to the dactylus of spearer-type raptorial appendage of stomatopods (mantis shrimp), which is similarly curved and equipped with hooked spines increasing in size distally (figure 8*e*) [74]. Traditionally spearing mantis shrimp appendages were thought to be used exclusively to ambush of agile soft prey, such as fish, while derived smashing mantis shrimp were thought to be exclusively durophagous [75]. However, recent isotopic analyses of the muscles of spearing mantis shrimps have revealed that they also consume hard-shelled prey, just in lower abundances than smashing mantis shrimps, which also consume some soft prey [76].

Although limited insight is possible into the functional morphology of isolated appendages of uncertain segmental affinities, do the similarities between the appendages of *Lihuacaris ferox* and those of stomatopods and *Sidneyia inexpectans* indicate that *Lihuacaris* was a predator capable of breaking biomineralized exoskeletons? A durophagous habit would make it a potential culprit for the repaired injuries of trilobites—examples of which have recently been described from the Stage 4 of South China [77]. To assess this possibility, it is necessary to consider the proximal portion of *Lihuacaris ferox* appendages, and the mechanics by which stomatopods and *Sidneyia* processed hard-shelled animals. Stomatopods generate the power required to break hard-shelled prey from a network of spring and latch structures, with a saddle-bearing merus, meral-V, carpus and propodus all working together to propel the dactyl at prey animals [78]. A faster propulsion of the dactyl creates the necessary power to break exoskeletons [55]. *Sidneyia inexpectans* appendages processed prey animals, including hard-shelled ones, using robust proximal gnathobases adjacent to their food groove [54,69]. There is no evidence that *Lihuacaris ferox* appendages have a complex network of spring and latch exoskeletal structures, thus stomatopod-style shell smashing is not supported. Likewise, there is no indication of robust molariform gnathobasic teeth in the base of *Lihuacaris ferox*, so *Sidneyia*-like shell crushing is not supported. It appears more likely that the distal element, with its numerous long hooked spines, instead performed a role in the capture of macroscopic non-biomineralized animals.

Traditionally, the trunk appendages of fuxianhuiids were thought to only have an ambulatory function because of their relatively simple construction and lack of specialized structures for feeding. However, recent work has revealed differentiated gnathobasic protopodites in the limbs of *Alacaris*

*mirabilis* and *Chengjiangocaris kunmingensis* [16] as well as the presence of well-developed subtriangular endites on the limbs of *Xiaocaris luoi* [38]. These appendicular specializations suggest that fuxianhuiids were able to consume macroscopic food items, rather than being restricted to deposit feeding as previously thought (e.g. [42]). The feeding ecology of fuxianhuiids is further informed by the presence of well-developed mid-gut glands in *Fuxianhuia protensa* [79], which have been regarded as anatomical adaptation for the storage and/or enzymatic digestion of food in Cambrian total-group euarthropods [8]. The isolated endopods of *Alacaris*? sp. from Guanshan represent the first record of highly differentiated fuxianhuiid appendages outside of the Xiaoshiba biota [16]. Their gnathobasic protopodites indicate that this organism was macrophagous and capable of food physical processing prior to ingestion. Determining whether the appendage of *Alacaris*? sp., if used for predation, were capable of breaking biomineralized exoskeletons would require a thorough biomechanical analysis, as the presence of a gnathobase alone is insufficient for inferring a durophagous habit [69].

# 4. Conclusion

The endemic taxa *Alacaris*? sp. and *Lihuacaris ferox* increase the known diversity of euarthropods in the Guanshan Biota. *Alacaris*? sp., the second fuxianhuiid reported from this exceptional biota, represents both the first occurrence of the Chengjiangocarididae in these beds and the youngest record of this fuxianhuiid family to date. The exact affinities of the new taxon *Lihuacaris ferox* are uncertain, but it most likely falls close to radiodonts in the euarthropod stem-lineage, or possibly fuxianhuiids as early members of Deuteropoda. Additional material of *Lihuacaris ferox* is critical for clarification of its relationships with other stem-group euarthropods and may potentially inform on the sequence of evolution of fundamental euarthropod characters.

Appendages of *Lihuacaris ferox* appear adapted for capturing non-biomineralized prey. This allows predatory habits to be inferred for this enigmatic taxon and suggests that radiodonts were not the only large predators in the Guanshan Biota. The appendages of *Alacaris*? sp. feature gnathobasic protopodite indicative of macrophagous feeding habits. The endemism of all these taxa probably reflects the relatively proximal shelf depositional settings of the Guanshan Konservat-Lagerstätte [34].

Data accessibility. The specimens studied here are accessioned at the Key Laboratory for Palaeobiology (YKLP), Yunnan University, Kunming, China. No additional data were generated during the course of the study. All data are presented in the main manuscript. No permissions were required for conducting this research.

Authors' contributions. J.Y., T.L. and X.-g.Z. designed the project. D.-g.J. and X.-g.Z. photographed material. D.-g.J. and X.-g.Z.. prepared figure 2, and S.P. prepared all other figures, with input from R.L.-A. and J.O.-H. D.-g.J., J.Y., T.L. and X.-g.Z. performed fieldwork, collected and prepared the material. S.P., R.L.-A. and J.O.-H. interpreted the data, and wrote the manuscript with input of all co-authors. All authors performed research, discussed and approved the final manuscript.

Competing interests. We declare we have no competing interests.

Funding. This study was supported by the National Natural Science Foundation of China (grant no. 41730318). S.P. acknowledges funding through an Alexander Agassiz Postdoctoral Fellowship (Museum of Comparative Zoology, Harvard University) and a Herchel Smith Postdoctoral Fellowship (University of Cambridge).

Acknowledgements. We thank C. Haug, J. Vannier and two anonymous referees for their comments and suggested improvements to the manuscript, and we thank K.S. Du, J.L. Du and W. Dong for their help with fieldwork.

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
