## [Peer Review File · Royal Society Open Science]

**New multipodomerous appendages of stem-group
euarthropods from the Cambrian (Stage 4) Guanshan
Konservat-Lagerstätte**

De-guang Jiao, Stephen Pates, Rudy Lerosey-Aubril, Javier Ortega-Hernández, Jie Yang,
Tian Lan and Xi-guang Zhang

Article citation details

R. Soc. open sci. **8**: 211134.
<http://dx.doi.org/10.1098/rsos.211134>

Review timeline

Original submission: 8 July 2021
Revised submission: 24 September 2021
Final acceptance: 4 October 2021

Note: Reports are unedited and appear as
submitted by the referee. The review history
appears in chronological order.

Review History

RSOS-211134.R0 (Original submission)

Review form: Reviewer 1

Is the manuscript scientifically sound in its present form?

Yes

Are the interpretations and conclusions justified by the results?

No

Is the language acceptable?

Yes

Do you have any ethical concerns with this paper?

No

Have you any concerns about statistical analyses in this paper?

No

Recommendation?

Major revision is needed (please make suggestions in comments)

Comments to the Author(s)

General comments:

The paper reports two novel forms of euarthropod appendages from the Cambrian Guanshan Biota of Yunnan, China. These appendages represent new fossil discoveries that expand our knowledge on the morphology, ecology, and evolution of Cambrian euarthropods. The predatory function of the new fuxianhuiid legs gains good support from data. The text is clearly written, and the figures are nicely illustrated.

I agree with the authors' interpretations of *Alacaris?* sp.. However, I have alternative ideas on the affinities of *Lihuacaris ferox* gen. and sp. nov.. While the authors conclude the appendage of *Lihuacaris ferox* as radiodont-like, below I list a number of points suggesting that the appendage is mostly likely a fuxianhuiid leg:

- (a) Tergite: The tergite associated with the appendage of *L. ferox* (YKLP 12438, Figure 6A) resembles those in *Guangweicaris spinatus*.
- (b) Length of appendage: The length of *L. ferox* appendage (ca. 28 mm) is normal for some fuxianhuiids, including *Chengjiangocaris*. So, this length is not characteristic of radiodont frontal appendages.
- (c) Gnathobase: The bottom left of Figure 6A shows a gnathobase, which resembles the one of *Alacaris?* sp. in Figure 4.
- (d) Linear groove or putative extensor muscle: The 'possible extensor muscle' ('em?' in Figures 6B, C, 7A), which starts from the base and ends in the claw, is almost identical to the 'linear groove' illustrated in *Alacaris?* sp. ('gr' in Figures 4, 5B, C; also shown in a number of other fuxianhuiids), but no similar structures are known from radiodont frontal appendages.
- (e) Kink: Radiodont frontal appendages bear their kink at the joint between shaft and podomere 1 of distal articulated region. On *L. ferox* appendage, the kink is situated between podomeres 3 to 5 (Figure 6B). The position of kink in *L. ferox* matches that in other fuxianhuiid legs, suggesting similar functional morphology (e.g., *Chengjiangocaris longiformis*, Figs. 13, 14 in Hou and Bergström, 1997).
- (f) Number of podomeres: The number of podomeres in *L. ferox* appendage (11 + claw) falls within the known range of podomere numbers in fuxianhuiid legs.
- (g) Arthrodial membranes: Although not emphasized in the literature, the triangular shape of arthrodial membranes is actually present in the legs of some fuxianhuiids (e.g., Fig. 13 in Hou and Bergström, 1997; Fig. 16.5 b in Hou et al., 2004; Fig. 4C in Chen et al., 2020; Fig. 2E1 in Wu et al., 2019).
- (h) Podomere shape: The homonomous rectangular podomeres are common in fuxianhuiids, whilst the podomeres of most radiodont frontal appendages are more trapezoid.
- (i) Endite: Most radiodont frontal appendages possess auxiliary spines on their endites, but auxiliary spines are absent in *L. ferox*. The endites of *L. ferox* (Figures 6B, C, 7B) are very similar to those in *Alacaris?* sp. (Figures 4, 5B).
- (j) Claw: The claw of *L. ferox* is certainly peculiar among the Cambrian euarthropods, but it has no counterparts in radiodonts.

Based on these points above, I insist that the *L. ferox* appendage is very likely a post-SPA leg of fuxianhuiid, and that the radiodont-like features of *L. ferox* appendage suggested in the manuscript are superficial. I would like to invite the authors to consider the information above, to incorporate useful points into the discussion of comparative morphology in the section 'Affinities of *Lihuacaris ferox*', and to revise their evolutionary discussion accordingly. The notion of *L.*

ferox appendage as radiodont-like may be weakened throughout the text. I would respect if the authors keep their discussion on the segmental affinities of *L. ferox* appendage.

Specific points:

Line 53. There is no lobopodian named as *Macrodictyon*. I suppose the authors might mean *Megadictyon*.

Lines 65-66. Aria et al. (2021, *Journal of Geological Society*) seems to be a better reference here for the post-tritocerebral hypothesis of fuxianhuiid specialized post-antennal appendages.

Lines 281-282. See point (b) above for the discussion on the length of appendage.

Lines 282-286. See point (h) above for the discussion on the shape of podomere.

Lines 286-289. See point (i) above for the discussion on the endite.

Lines 289-292. See point (d) above for the discussion on the linear groove or putative extensor muscle.

Lines 295-374. I would like to invite the authors to revise these paragraphs by considering the potentially helpful points (a-j) above.

Lines 376-437. The authors may revise this section 'Position of *Lihuacaris ferox* in the euarthropod stem lineage' if they change their ideas on the affinity of *Lihuacaris ferox*.

Lines 489-490. The conclusion on the possible affinities of *L. ferox* may require revision, as its appendage shows many fuxianhuiid features.

Figure 1B. The segmental affinity of radiodont frontal appendages was coded as 'questionable' ('?'; protocerebral or deutocerebral) in the morphological matrix of Zeng et al. (2020). However, regardless of the uncertainty pointed out in the original reference, the illustration in panel B simply locates the frontal appendage at a protocerebral position. A correction recognizing the uncertainty needs to be made on panel B.

Figure 6. The abbreviations for tergite do not match in the panels ('te') and caption ('t'). The explanation for abbreviation 'gn' is missing in the caption.

References:

Aria C., Zhao F., Zhu M. 2021 Fuxianhuiids are mandibulates and share affinities with total-group Myriapoda. *Journal of the Geological Society*, jgs2020-2246. (doi:10.1144/jgs2020-246).

Chen H., Legg D.A., Zhai D., Liu Y., Hou X. 2020 New data on the anatomy of fuxianhuiid arthropod *Guangweicaris spinatus* from the lower Cambrian Guanshan Biota, Yunnan, China. *Acta Palaeontologica Polonica* 65(1), 139-148.

Hou X., Bergström J. 1997 Arthropods of the lower Cambrian Chengjiang fauna, southwest China. *Fossils and Strata* 45, 1-116.

Hou X., Aldridge R.J., Bergström J., Siveter D.J., Siveter D.J., Feng X. 2004 *The Cambrian fossils of Chengjiang, China: the flowering of early animal life*, Blackwell Science Ltd; 233 p.

Wu Y., Liu J. 2019 Anatomy and relationships of the fuxianhuiid euarthropod *Guangweicaris* from the early Cambrian Guanshan Biota in Kunming, Yunnan, Southwest China revisited. *Acta Palaeontologica Polonica* 64(3), 543-548.

Zeng H., Zhao F., Niu K., Zhu M., Huang D. 2020 An early Cambrian euarthropod with radiodont-like raptorial appendages. *Nature* 588(7836), 101-105. (doi:10.1038/s41586-020-2883-7).

Review form: Reviewer 2

Is the manuscript scientifically sound in its present form?

Yes

Are the interpretations and conclusions justified by the results?

Yes

Is the language acceptable?

Yes

Do you have any ethical concerns with this paper?

No

Have you any concerns about statistical analyses in this paper?

No

Recommendation?

Accept with minor revision (please list in comments)

Comments to the Author(s)

Comments:

This manuscript by Jiao and his colleagues described two new arthropods, *Lihuacaris ferox* gen. et sp. nov. and *Alacaris?* sp. from Cambrian Guanshan deposit, in combination with a discussion on the evolution of euarthropod characters and the feeding strategy.

It is no doubt that *Lihuacaris ferox* represents an interesting new species, whose appendage shows unique combined features of Radiodonta and Deuteropoda. The morphology of the appendage, especially its unique spinose distal element, expanded the morphospace of the euarthropod limbs. I agree with authors that the new taxon most likely falls close to radiodonts in the Euarthropoda, though lack of the whole-body information precludes a precise systematic assignment within the Arthropoda. I am also happy to see various evolutionary scenarios suggested by authors based on the different positions of *Lihuacaris* within the total group of Euarthropoda.

For the latter taxon, *Alacaris?* sp., the authors identified the specimens as isolated fuxianhuiid endopods, and assign into family Chengjiangacaridae. But I am hesitating to define *Alacaris?* sp. fossils as the endopod. The endite described in the manuscript, marked as "en" particular in Figure 4 and 5 show striking similarities with the pleural spine of the dorsal tergite, particular that in *Fuxianhuiida*. The size of structure sounds too big to be the endopod of fuxianhuiids. Because in the group of *Fuxianhuiida*, each dorsal tergite covers multiple appendages. However, the "height of the podomere" of the endopod is 7 mm (see Line 47), that is three times wider than the width of the tergite closely preserved to it (Fig. 4). Furthermore, the evidence of the teeth and slender spines attached to the hypothetical protopod is not very convincing. I found that the so called protopod part, preserved on the upper bedding and covered some segments (see Figs. 3 and 4), is much more similar to the dorsal exoskeleton of euarthropod.

I have not seen the fossils in person, so it has the possibility of being the endopod. So, it is strongly recommended a double check of *Alacaris?* sp. specimens, just in case. But the importance of this manuscript won't be affected by the definition of the endopod in *Alacaris?* sp. In addition, I would suggest to use "prodopod" instead of "prodopodite" when describe a non-crustacean arthropod.

Some minor mistakes:

Line 63: should be [e.g. 18,19,20]

Line 134-149: in this part, the number of the specimens described in the text was mismatched with the specimen number in the Figures 3-5. For example, in line 136, the authors mentioned the specimen 12434 (Fig. 4), while the specimen number is 12435 in the figure caption. Please check this part.

Line 201: add space between et and sp.

In summary, it is a well written manuscript. I recommend publication in Royal Society Open Science after the minor revision.

Review form: Reviewer 3 (Caroline Haug)

Is the manuscript scientifically sound in its present form?

Yes

Are the interpretations and conclusions justified by the results?

Yes

Is the language acceptable?

Yes

Do you have any ethical concerns with this paper?

No

Have you any concerns about statistical analyses in this paper?

No

Recommendation?

Accept with minor revision (please list in comments)

Comments to the Author(s)

The authors describe in this paper two new species of Euarthropoda, one of them being the youngest occurrence of Chenjiangocaridae. The material is interesting in the context of arthropod evolution and merits publication. The text and the figures are in general of good quality, but I have some comments on some language inconsistencies and on improvements/amendments to the figures (see below). I recommend publication after minor revision.

20: diverse can be used in different ways, better use species-rich

20: As the rank height for a group is rather arbitrary, "the most diverse animal phylum" has no true meaning in a scientific way. I suggest to rephrase the sentence.

22: Definitions are done in humanities, I suggest “characterise” instead.

22: Though often used, “body plan” implies a planner, which is problematic in the days of Intelligent Design and other creationists, better rephrase.

26: here and in other places: try to avoid Linnean ranks (e.g. also in the Systematic palaeontology section)

28: here and in other places: Use also the species name if you are not explicitly talking about the genus only.

29: “a hypertrophied”, not an, as the h is voiced

50 ff.: Sentence starting with “Among” is erroneous grammatically, please check.

56: Better phrase “modern representatives of Euarthropoda”

56: As synapomorphies are new characters shared by two groups, the sentence is phrased a bit unfortunate. If these characters would be autapomorphies of Euarthropoda, that would not fit with their occurrence also in radiodonts. Most probably, these characters already occurred earlier and are hence plesiomorphies, but characteristic for Euarthropoda (and radiodonts). Please rephrase.

171: Here it becomes especially apparent that deleting “family” would be easy and even improve the readability.

175: Here, taxon can be changed to species, which provides more information (check also in other places).

210: caris is not Latin, but Greek

273: I recommend to restrict segment to body segments, for appendages it is better to use elements (check also in other places).

278: Radiodonta as group name should be used as singular (though often plural is used by many authors).

322: “a hypertrophied”, not an, as the h is voiced

350: typo in Lihuacaris

384: Lihuacaris should not be italicised as the rest of the header is (also check other headers)

388 ff.: The function of these appendages does not depend on their segmental affiliation, but the sentence creates this impression, please rephrase.

404/405: The appendages for sure do not arise from the deutocerebrum, but they may have been innervated by the deutocerebrum or may arise from the deutocerebral segment, please rephrase.

405: Not Lihuacaris is similar to the first appendages, but its appendages are similar. Be careful with the references.

421 ff: Sentence incomplete, please check.

833: typo in megacheirans

Fig. 1: The appendages should all be positioned in the same orientation (e.g. spines to the bottom) to allow better comparability.

Fig. 2. The font is in most cases too small, hardly readable (please check also journal guidelines). Better rearrange the figure to more square-shape, then you can make everything a bit larger.

Figs. 3–7: For future photographs I recommend to use cross-polarised light (polarisation filter in front of camera lens, perpendicular polarisation filters in front of light source) as that strongly enhances the contrast between fossils and matrix. For the current photos, I recommend to optimise the histogram for a better contrast.

Fig. 8: There is a reconstruction of *Lihuacaris ferox*, but I would also like to see one of *Alacaris* ?, even if incomplete.

Fig. 9: You need to explain what the grey lines in the phylograms represent. I also think that it would be necessary to name the presumed outgroup in B to be able to place the apomorphy “arthropodized limbs” at that position.

Carolin Haug, LMU Munich

Review form: Reviewer 4 (Jean Vannier)

Is the manuscript scientifically sound in its present form?

Yes

Are the interpretations and conclusions justified by the results?

Yes

Is the language acceptable?

Yes

Do you have any ethical concerns with this paper?

No

Have you any concerns about statistical analyses in this paper?

No

Recommendation?

Accept with minor revision (please list in comments)

Comments to the Author(s)

This is a good MS based on very accurate descriptions of a fragmentary but well-preserved arthropod material from the Guanshan Lagerstätte, with interesting discussions on euarthropod early evolution and autecology. It is well written and easy to follow. I would suggest to slightly re-arrange/shorten the paragraph on the assumed feeding mode of *Lihuacaris* (see below).

Figures and diagrams are of excellent quality. Figure 1 is particularly useful to those readers who

may not be familiar with current debates on euarthropod evolution. I would be happy to recommend this MS for publication in this journal.

Line 88- Please give some general information on the faunal composition of the Guanshan Biota and especially how it differs from other Lagerstätten (e.g. Chengjiang). Why do you think that shallower environments resulted in radiodont endemism? Do we know the approximate age of this formation (e.g. biozonations)?

Line 125 +

The description of *Alacaris*? sp. is based on two isolated appendages. I am just wondering whether more abundant fragments of other body features could be found in the Guanshan Biota, that would more strongly support its affiliation to *Alacaris*? Is this biota characterized by disarticulated arthropod fragments?

Since your figures look quite large, why not figure a complete specimen of *Alacaris* (from a different locality) in one of them. That would help readers locate the isolated appendages shown in the MS. Perhaps ask a colleague to provide such illustration?

Line 165: again here (see above) a picture of *Guangweicaris* would be useful.

Line 169: Perhaps what you have here is the isolated appendages of large specimen of *Guangweicaris* (see number of podomeres). It would not be surprising that large specimens had more chance to be disarticulated by mudflows or currents.

Does *Xiaocaris* really lack gnathobasic protopodites? Perhaps they are concealed by other features, Please check.

Line 177- That's why I made the remark above.

Line 186- Please, be careful here. Can you estimate the approx. size of the arthropod that bore the isolated appendage? Comparisons should be based on specimens with a comparable size.

Line 201. *Lihuacaris ferox*

This new species is mainly based on two isolated appendages (one of them being incomplete). I agree that at least one appendage provides clear morphological information. However, isn't it a bit risky to make a new genus and species with such a fragmentary material?

Line 233-234

Not very clear here. Is it the carapace ornament of the bivalved arthropod?

Line 237

I am not sure that the holotype of a new species can be an isolated appendage. It would mean that this new taxon is defined of a very small percentage of its whole anatomy. We don't even know the size of the whole animal. Please check.

Line 281

It seems that other radiodonts occur in the same biota. Detailed comparisons should be made with them (again images would be useful).

Line 290

Do you have muscle remains?

Line 307

It is clear that this appendage resembles that of numerous radiodont species. What makes it so special is not completely clear to me. What do you mean by the “level of separation”. This character may result from post-mortem decay of muscles

Line 318

Do you mean that the appendages of Lihuacaris had a special kind of highly flexible articulation?

Line 324

There is no speculation here. the terminal “claw” looks like a fused element (no podomere boundaries visible)

Line 333

Yes, interesting hypothesis.

Line 384

Yes, it is one option to be discussed. However, since your appendages are isolated, we don't know to which part of the head they were attached. I find this part (>> 438) a bit speculative?

Line 402

Same here. One of the three options is correct but additional fossil evidence (relation to head) critical to the discussion is lacking.

Line 441

These appendages are quite large and bear a strong terminal “claws”. Is it enough to make them predators? I see no obvious prehensile features unless the appendage. Sidneyia could maintain small prey along its ventral part and masticate them by using strong gnathobases. I see no such hard elements in (e.g.) Fig. 8.

Line 444

Comparisons with mantis-shrimps seems to be a bit far-fetched. I don't find them very convincing. I see no raptorial elements in the appendage of Lihuacaris unless they could fold up ventrally. If possible provide a diagram (add to Fig. 8 ?) showing both the “resting” and “folded” position. You could do the same for the other appendages shown in Fig. 8. It would greatly help readers understand the function of appendages.

Line 452

Sidneyia and stomatopods are two different types of animal in terms of predation. Sidneyia was most probably a slow animal feeding on epibenthic prey. Mantis shrimps are extremely fast hunters.

Line 452 +

You are embarking into detailed discussions on the feeding mode of Lihuacaris. Think that these discussions are based on a couple of isolated appendages only. It might be a bit risky to discuss the feeding mode of an animal on the basis of such a fragmentary material. We know nothing about its size, morphology of other appendages, gut, etc...Please improve this part.

Line 464. You see... (impossible)

Line 465. Idem

Perhaps start this paragraph by saying that unfortunately very little is known of the actual functional morphology of the animal.

Line 468

Yes, I find it interesting to mention the feeding habits of Fuxianhuia. This is a well- documented taxon with plenty of excellently preserved specimens. Unfortunately again, your specimens are fragmentary. It is frustrating in some way. Perhaps, try to avoid “what-if-they-had” long discussions.

Line 483

This statement is perfectly correct.

Although the present material is fragmentary, new studies of the Guanshan Biota might reveal unknown aspects of arthropod evolution. Try to stress on the importance of this biota (its potential interest, environmental context, etc.)- Also in the abstract.

Jean Vannier

Decision letter (RSOS-211134.R0)

Dear Dr Ortega-Hernández

The Editors assigned to your paper RSOS-211134 "New multipodomerous appendages of stem group euarthropods from the Cambrian (Stage 4) Guanshan Konservat-Lagerstätte" have now received comments from reviewers and would like you to revise the paper in accordance with the reviewer comments and any comments from the Editors. Please note this decision does not guarantee eventual acceptance.

Please submit your revised manuscript and required files (see below) no later than 21 days from today's (ie 18-Aug-2021) date. Note: the ScholarOne system will 'lock' if submission of the revision is attempted 21 or more days after the deadline. If you do not think you will be able to meet this deadline please contact the editorial office immediately.

on behalf of Professor Allison Daley (Associate Editor) and Kevin Padian (Subject Editor)
 openscience@royalsociety.org

Associate Editor Comments to Author (Professor Allison Daley):

The reviews are very positive about the descriptive nature and the quality of this writing in this manuscript, and applaud the way that multiple evolutionary scenarios are explored. Two of the detailed reviews suggest that the authors reconsider the affinity of both the new taxa described in this paper. Reviewer 1 suggests a fuxianhuiid affinity for *Lihuacaris ferox* should be considered, and the arguments presented are convincing enough that I suggest the authors add some consideration of the fuxianhuiid characteristics described by Reviewer 1 to the discussion of the affinity of this strange appendage. For the material identified as *Alacaris?* sp., reviewer 2 raised some valid points about the identification as a fuxianhuiid endopod, which should also be considered by the authors in a revision of the manuscript. Reviewer 3 has numerous small suggestions that should be considered, especially the suggested modifications to the figures. I recommend a major revision of the manuscript with a consideration of these points raised by the reviewers.

Reviewer comments to Author:

Reviewer: 1

Comments to the Author(s)

General comments:

The paper reports two novel forms of euarthropod appendages from the Cambrian Guanshan Biota of Yunnan, China. These appendages represent new fossil discoveries that expand our knowledge on the morphology, ecology, and evolution of Cambrian euarthropods. The predatory function of the new fuxianhuiid legs gains good support from data. The text is clearly written, and the figures are nicely illustrated.

I agree with the authors' interpretations of *Alacaris?* sp.. However, I have alternative ideas on the affinities of *Lihuacaris ferox* gen. and sp. nov.. While the authors conclude the appendage of *Lihuacaris ferox* as radiodont-like, below I list a number of points suggesting that the appendage is mostly likely a fuxianhuiid leg:

- (a) Tergite: The tergite associated with the appendage of *L. ferox* (YKLP 12438, Figure 6A) resembles those in *Guangweicaris spinatus*.
- (b) Length of appendage: The length of *L. ferox* appendage (ca. 28 mm) is normal for some fuxianhuiids, including *Chengjiangocaris*. So, this length is not characteristic of radiodont frontal appendages.
- (c) Gnathobase: The bottom left of Figure 6A shows a gnathobase, which resembles the one of *Alacaris?* sp. in Figure 4.
- (d) Linear groove or putative extensor muscle: The 'possible extensor muscle' ('em?' in Figures 6B, C, 7A), which starts from the base and ends in the claw, is almost identical to the 'linear groove' illustrated in *Alacaris?* sp. ('gr' in Figures 4, 5B, C; also shown in a number of other fuxianhuiids), but no similar structures are known from radiodont frontal appendages.

(e) Kink: Radiodont frontal appendages bear their kink at the joint between shaft and podomere 1 of distal articulated region. On *L. ferox* appendage, the kink is situated between podomeres 3 to 5 (Figure 6B). The position of kink in *L. ferox* matches that in other fuxianhuiid legs, suggesting similar functional morphology (e.g., *Chengjiangocaris longiformis*, Figs. 13, 14 in Hou and Bergström, 1997).

(f) Number of podomeres: The number of podomeres in *L. ferox* appendage (11 + claw) falls within the known range of podomere numbers in fuxianhuiid legs.

(g) Arthrodistal membranes: Although not emphasized in the literature, the triangular shape of arthrodistal membranes is actually present in the legs of some fuxianhuiids (e.g., Fig. 13 in Hou and Bergström, 1997; Fig. 16.5 b in Hou et al., 2004; Fig. 4C in Chen et al., 2020; Fig. 2E1 in Wu et al., 2019).

(h) Podomere shape: The homonomous rectangular podomeres are common in fuxianhuiids, whilst the podomeres of most radiodont frontal appendages are more trapezoid.

(i) Endite: Most radiodont frontal appendages possess auxiliary spines on their endites, but auxiliary spines are absent in *L. ferox*. The endites of *L. ferox* (Figures 6B, C, 7B) are very similar to those in *Alacaris?* sp. (Figures 4, 5B).

(j) Claw: The claw of *L. ferox* is certainly peculiar among the Cambrian euarthropods, but it has no counterparts in radiodonts.

Based on these points above, I insist that the *L. ferox* appendage is very likely a post-SPA leg of fuxianhuiid, and that the radiodont-like features of *L. ferox* appendage suggested in the manuscript are superficial. I would like to invite the authors to consider the information above, to incorporate useful points into the discussion of comparative morphology in the section 'Affinities of *Lihuacaris ferox*', and to revise their evolutionary discussion accordingly. The notion of *L. ferox* appendage as radiodont-like may be weakened throughout the text. I would respect if the authors keep their discussion on the segmental affinities of *L. ferox* appendage.

Specific points:

Line 53. There is no lobopodian named as *Macrodictyon*. I suppose the authors might mean *Megadictyon*.

Lines 65-66. Aria et al. (2021, *Journal of Geological Society*) seems to be a better reference here for the post-tritocerebral hypothesis of fuxianhuiid specialized post-antennal appendages.

Lines 281-282. See point (b) above for the discussion on the length of appendage.

Lines 282-286. See point (h) above for the discussion on the shape of podomere.

Lines 286-289. See point (i) above for the discussion on the endite.

Lines 289-292. See point (d) above for the discussion on the linear groove or putative extensor muscle.

Lines 295-374. I would like to invite the authors to revise these paragraphs by considering the potentially helpful points (a-j) above.

Lines 376-437. The authors may revise this section 'Position of *Lihuacaris ferox* in the euarthropod stem lineage' if they change their ideas on the affinity of *Lihuacaris ferox*.

Lines 489-490. The conclusion on the possible affinities of *L. ferox* may require revision, as its appendage shows many fuxianhuiid features.

Figure 1B. The segmental affinity of radiodont frontal appendages was coded as ‘questionable’ (‘?’; protocerebral or deutocerebral) in the morphological matrix of Zeng et al. (2020). However, regardless of the uncertainty pointed out in the original reference, the illustration in panel B simply locates the frontal appendage at a protocerebral position. A correction recognizing the uncertainty needs to be made on panel B.

Figure 6. The abbreviations for tergite do not match in the panels (‘te’) and caption (‘t’). The explanation for abbreviation ‘gn’ is missing in the caption.

References:

Aria C., Zhao F., Zhu M. 2021 Fuxianhuidids are mandibulates and share affinities with total-group Myriapoda. *Journal of the Geological Society*, jgs2020-2246. (doi:10.1144/jgs2020-246).

Chen H., Legg D.A., Zhai D., Liu Y., Hou X. 2020 New data on the anatomy of fuxianhuidid arthropod *Guangweicaris spinatus* from the lower Cambrian Guanshan Biota, Yunnan, China. *Acta Palaeontologica Polonica* 65(1), 139-148.

Hou X., Bergström J. 1997 Arthropods of the lower Cambrian Chengjiang fauna, southwest China. *Fossils and Strata* 45, 1-116.

Hou X., Aldridge R.J., Bergström J., Siveter D.J., Siveter D.J., Feng X. 2004 *The Cambrian fossils of Chengjiang, China: the flowering of early animal life*, Blackwell Science Ltd; 233 p.

Wu Y., Liu J. 2019 Anatomy and relationships of the fuxianhuidid euarthropod *Guangweicaris* from the early Cambrian Guanshan Biota in Kunming, Yunnan, Southwest China revisited. *Acta Palaeontologica Polonica* 64(3), 543-548.

Zeng H., Zhao F., Niu K., Zhu M., Huang D. 2020 An early Cambrian euarthropod with radiodont-like raptorial appendages. *Nature* 588(7836), 101-105. (doi:10.1038/s41586-020-2883-7).

Reviewer: 2

Comments to the Author(s)

Comments:

This manuscript by Jiao and his colleagues described two new arthropods, *Lihuacaris ferox* gen. et sp. nov. and *Alacaris?* sp. from Cambrian Guanshan deposit, in combination with a discussion on the evolution of euarthropod characters and the feeding strategy.

It is no doubt that *Lihuacaris ferox* represents an interesting new species, whose appendage shows unique combined features of Radiodonta and Deuteropoda. The morphology of the appendage, especially its unique spinose distal element, expanded the morphospace of the euarthropod limbs. I agree with authors that the new taxon most likely falls close to radiodonts in the Euarthropoda, though lack of the whole-body information precludes a precise systematic assignment within the Arthropoda. I am also happy to see various evolutionary scenarios suggested by authors based on the different positions of *Lihuacaris* within the total group of Euarthropoda.

For the latter taxon, *Alacaris?* sp., the authors identified the specimens as isolated fuxianhuidid endopods, and assign into family Chengjiangacaridae. But I am hesitating to define *Alacaris?* sp. fossils as the endopod. The endite described in the manuscript, marked as “en” particular in Figure 4 and 5 show striking similarities with the pleural spine of the dorsal tergite, particular that in Fuxianhuidida. The size of structure sounds too big to be the endopod of fuxianhuidids. Because in the group of Fuxianhuidida, each dorsal tergite covers multiple appendages. However, the “height of the podomere” of the endopod is 7 mm (see Line 47), that is three times wider than

the width of the tergite closely preserved to it (Fig. 4). Furthermore, the evidence of the teeth and slender spines attached to the hypothetical protopod is not very convincing. I found that the so called protopod part, preserved on the upper bedding and covered some segments (see Figs. 3 and 4), is much more similar to the dorsal exoskeleton of euarthropod.

I have not seen the fossils in person, so it has the possibility of being the endopod. So, it is strongly recommended a double check of *Alacaris?* sp. specimens, just in case. But the importance of this manuscript won't be affected by the definition of the endopod in *Alacaris?* sp. In addition, I would suggest to use "prodopod" instead of "prodopodite" when describe a non-crustacean arthropod.

Some minor mistakes:

Line 63: should be [e.g. 18,19,20]

Line 134-149: in this part, the number of the specimens described in the text was mismatched with the specimen number in the Figures 3-5. For example, in line 136, the authors mentioned the specimen 12434 (Fig. 4), while the specimen number is 12435 in the figure caption. Please check this part.

Line 201: add space between et and sp.

In summary, it is a well written manuscript. I recommend publication in Royal Society Open Science after the minor revision.

Reviewer: 3

Comments to the Author(s)

The authors describe in this paper two new species of Euarthropoda, one of them being the youngest occurrence of Chenjiangocaridae. The material is interesting in the context of arthropod evolution and merits publication. The text and the figures are in general of good quality, but I have some comments on some language inconsistencies and on improvements/amendments to the figures (see below). I recommend publication after minor revision.

20: diverse can be used in different ways, better use species-rich

20: As the rank height for a group is rather arbitrary, "the most diverse animal phylum" has no true meaning in a scientific way. I suggest to rephrase the sentence.

22: Definitions are done in humanities, I suggest "characterise" instead.

22: Though often used, "body plan" implies a planner, which is problematic in the days of Intelligent Design and other creationists, better rephrase.

26: here and in other places: try to avoid Linnean ranks (e.g. also in the Systematic palaeontology section)

28: here and in other places: Use also the species name if you are not explicitly talking about the genus only.

29: "a hypertrophied", not an, as the h is voiced

50 ff.: Sentence starting with "Among" is erroneous grammatically, please check.

56: Better phrase “modern representatives of Euarthropoda”

56: As synapomorphies are new characters shared by two groups, the sentence is phrased a bit unfortunate. If these characters would be autapomorphies of Euarthropoda, that would not fit with their occurrence also in radiodonts. Most probably, these characters already occurred earlier and are hence plesiomorphies, but characteristic for Euarthropoda (and radiodonts). Please rephrase.

171: Here it becomes especially apparent that deleting “family” would be easy and even improve the readability.

175: Here, taxon can be changed to species, which provides more information (check also in other places).

210: caris is not Latin, but Greek

273: I recommend to restrict segment to body segments, for appendages it is better to use elements (check also in other places).

278: Radiodonta as group name should be used as singular (though often plural is used by many authors).

322: “a hypertrophied”, not an, as the h is voiced

350: typo in Lihuacaris

384: Lihuacaris should not be italicised as the rest of the header is (also check other headers)

388 ff.: The function of these appendages does not depend on their segmental affiliation, but the sentence creates this impression, please rephrase.

404/405: The appendages for sure do not arise from the deutocerebrum, but they may have been innervated by the deutocerebrum or may arise from the deutocerebral segment, please rephrase.

405: Not Lihuacaris is similar to the first appendages, but its appendages are similar. Be careful with the references.

421 ff: Sentence incomplete, please check.

833: typo in megacheirans

Fig. 1: The appendages should all be positioned in the same orientation (e.g. spines to the bottom) to allow better comparability.

Fig. 2. The font is in most cases too small, hardly readable (please check also journal guidelines). Better rearrange the figure to more square-shape, then you can make everything a bit larger.

Figs. 3–7: For future photographs I recommend to use cross-polarised light (polarisation filter in front of camera lens, perpendicular polarisation filters in front of light source) as that strongly enhances the contrast between fossils and matrix. For the current photos, I recommend to optimise the histogram for a better contrast.

Fig. 8: There is a reconstruction of *Lihuacaris ferox*, but I would also like to see one of *Alacaris* ?, even if incomplete.

Fig. 9: You need to explain what the grey lines in the phylograms represent. I also think that it would be necessary to name the presumed outgroup in B to be able to place the apomorphy “arthropodized limbs” at that position.

Carolin Haug, LMU Munich

Reviewer: 4

Comments to the Author(s)

This is a good MS based on very accurate descriptions of a fragmentary but well-preserved arthropod material from the Guanshan Lagerstätte, with interesting discussions on euarthropod early evolution and autecology. It is well written and easy to follow. I would suggest to slightly re-arrange/shorten the paragraph on the assumed feeding mode of *Lihuacaris* (see below). Figures and diagrams are of excellent quality. Figure 1 is particularly useful to those readers who may not be familiar with current debates on euarthropod evolution. I would be happy to recommend this MS for publication in this journal.

Line 88- Please give some general information on the faunal composition of the Guanshan Biota and especially how it differs from other Lagerstätten (e.g. Chengjiang). Why do you think that shallower environments resulted in radiodont endemism? Do we know the approximate age of this formation (e.g. biozonations)?

Line 125 +

The description of *Alacaris*? sp. is based on two isolated appendages. I am just wondering whether more abundant fragments of other body features could be found in the Guanshan Biota, that would more strongly support its affiliation to *Alacaris*? Is this biota characterized by disarticulated arthropod fragments?

Since your figures look quite large, why not figure a complete specimen of *Alacaris* (from a different locality) in one of them. That would help readers locate the isolated appendages shown in the MS. Perhaps ask a colleague to provide such illustration?

Line 165: again here (see above) a picture of *Guangweicaris* would be useful.

Line 169: Perhaps what you have here is the isolated appendages of large specimen of *Guangweicaris* (see number of podomeres). It would not be surprising that large specimens had more chance to be disarticulated by mudflows or currents.

Does *Xiaocaris* really lack gnathobasic protopodites? Perhaps they are concealed by other features, Please check.

Line 177- That's why I made the remark above.

Line 186- Please, be careful here. Can you estimate the approx. size of the arthropod that bore the isolated appendage? Comparisons should be based on specimens with a comparable size.

Line 201. *Lihuacaris ferox*

This new species is mainly based on two isolated appendages (one of them being incomplete). I agree that at least one appendage provides clear morphological information. However, isn't it a bit risky to make a new genus and species with such a fragmentary material?

Line 233-234

Not very clear here. Is it the carapace ornament of the bivalved arthropod?

Line 237

I am not sure that the holotype of a new species can be an isolated appendage. It would mean that this new taxon is defined of a very small percentage of its whole anatomy. We don't even know the size of the whole animal. Please check.

Line 281

It seems that other radiodonts occur in the same biota. Detailed comparisons should be made with them (again images would be useful).

Line 290

Do you have muscle remains?

Line 307

It is clear that this appendage resembles that of numerous radiodont species. What makes it so special is not completely clear to me. What do you mean by the "level of separation". This character may result from post-mortem decay of muscles

Line 318

Do you mean that the appendages of *Lihuacaris* had a special kind of highly flexible articulation?

Line 324

There is no speculation here. the terminal "claw" looks like a fused element (no podomere boundaries visible)

Line 333

Yes, interesting hypothesis.

Line 384

Yes, it is one option to be discussed. However, since your appendages are isolated, we don't know to which part of the head they were attached. I find this part (>> 438) a bit speculative?

Line 402

Same here. One of the three options is correct but additional fossil evidence (relation to head) critical to the discussion is lacking.

Line 441

These appendages are quite large and bear a strong terminal "claws". Is it enough to make them predators? I see no obvious prehensile features unless the appendage. *Sidneyia* could maintain small prey along its ventral part and masticate them by using strong gnathobases. I see no such hard elements in (e.g.) Fig. 8.

Line 444

Comparisons with mantis-shrimps seems to be a bit far-fetched. I don't find them very convincing. I see no raptorial elements in the appendage of *Lihuacaris* unless they could fold up ventrally. If possible provide a diagram (add to Fig. 8 ?) showing both the "resting" and "folded" position. You could do the same for the other appendages shown in Fig. 8. It would greatly help readers understand the function of appendages.

Line 452

Sidneyia and stomatopods are two different types of animal in terms of predation. Sidneyia was most probably a slow animal feeding on epibenthic prey. Mantis shrimps are extremely fast hunters.

Line 452 +

You are embarking into detailed discussions on the feeding mode of Lihucaris. Think that these discussions are based on a couple of isolated appendages only. It might be a bit risky to discuss the feeding mode of an animal on the basis of such a fragmentary material. We know nothing about its size, morphology of other appendages, gut, etc...Please improve this part.

Line 464. You see... (impossible)

Line 465. Idem

Perhaps start this paragraph by saying that unfortunately very little is known of the actual functional morphology of the animal.

Line 468

Yes, I find it interesting to mention the feeding habits of Fuxianhuia. This is a well- documented taxon with plenty of excellently preserved specimens. Unfortunately again, your specimens are fragmentary. It is frustrating in some way. Perhaps, try to avoid "what-if-they-had" long discussions.

Line 483

This statement is perfectly correct.

Although the present material is fragmentary, new studies of the Guanshan Biota might reveal unknown aspects of arthropod evolution. Try to stress on the importance of this biota (its potential interest, environmental context, etc.)- Also in the abstract.

Jean Vannier

===PREPARING YOUR MANUSCRIPT===

===PREPARING YOUR REVISION IN SCHOLARONE===

Author's Response to Decision Letter for (RSOS-211134.R0)

See Appendix A.

Decision letter (RSOS-211134.R1)

Dear Dr Ortega-Hernández,

It is a pleasure to accept your manuscript entitled "New multipodomorous appendages of stem group euarthropods from the Cambrian (Stage 4) Guanshan Konservat-Lagerstätte" in its current form for publication in Royal Society Open Science. The comments of the reviewer(s) who reviewed your manuscript are included at the foot of this letter.

Kind regards,
Royal Society Open Science Editorial Office

on behalf of Professor Allison Daley (Associate Editor) and Kevin Padian (Subject Editor)
openscience@royalsociety.org

Appendix A

Editor and reviewer comments

on behalf of Professor Allison Daley (Associate Editor) and Kevin Padian (Subject Editor)
openscience@royalsociety.org

Associate Editor Comments to Author (Professor Allison Daley):

The reviews are very positive about the descriptive nature and the quality of this writing in this manuscript, and applaud the way that multiple evolutionary scenarios are explored. Two of the detailed reviews suggest that the authors reconsider the affinity of both the new taxa described in this paper. Reviewer 1 suggests a fuxianhuiid affinity for *Lihuacaris ferox* should be considered, and the arguments presented are convincing enough that I suggest the authors add some consideration of the fuxianhuiid characteristics described by Reviewer 1 to the discussion of the affinity of this strange appendage. For the material identified as *Alacaris?* sp., reviewer 2 raised some valid points about the identification as a fuxianhuiid endopod, which should also be considered by the authors in a revision of the manuscript. Reviewer 3 has numerous small suggestions that should be considered, especially the suggested modifications to the figures. I recommend a major revision of the manuscript with a consideration of these points raised by the reviewers.

We thank the Associate Editor and the four reviewers for their detailed comments and positive response to our manuscript. We have expanded our manuscript to include some of the additional possible affinities for the specimens we describe, and have adjusted some of the figures as suggested.

Detailed responses to the reviewer comments are outlined point by point below, and a tracked changes version of the manuscript is also provided. We feel that our manuscript is now ready for publication. We truly appreciate the time taken by the Associate Editor and the four reviewers to provide constructive and detailed criticism of our work.

Reviewer comments to Author:

Reviewer: 1

Comments to the Author(s)

General comments:

The paper reports two novel forms of euarthropod appendages from the Cambrian Guanshan Biota of Yunnan, China. These appendages represent new fossil discoveries that expand our knowledge on the morphology, ecology, and evolution of Cambrian euarthropods. The predatory function of the new fuxianhuid legs gains good support from data. The text is clearly written, and the figures are nicely illustrated.

We thank the reviewer for their positive feedback. We appreciate the time taken to put together the review, and to offer an alternative explanation for the affinities of *Lihuacaris ferox*. While we do not agree with this interpretation (see below), the referees constructive comments have helped us to strengthen the manuscript.

I agree with the authors' interpretations of *Alacaris?* sp.. However, I have alternative ideas on the affinities of *Lihuacaris ferox* gen. and sp. nov.. While the authors conclude the appendage of *Lihuacaris ferox* as radiodont-like, below I list a number of points suggesting that the appendage is mostly likely a fuxianhuid leg:

- (a) Tergite: The tergite associated with the appendage of *L. ferox* (YKLP 12438, Figure 6A) resembles those in *Guangweicaris spinatus*.
- (b) Length of appendage: The length of *L. ferox* appendage (ca. 28 mm) is normal for some fuxianhuids, including *Chengjiangocaris*. So, this length is not characteristic of radiodont frontal appendages.
- (c) Gnathobase: The bottom left of Figure 6A shows a gnathobase, which resembles the one of *Alacaris?* sp. in Figure 4.
- (d) Linear groove or putative extensor muscle: The 'possible extensor muscle' ('em?' in Figures 6B, C, 7A), which starts from the base and ends in the claw, is almost identical to the 'linear groove' illustrated in *Alacaris?* sp. ('gr' in Figures 4, 5B, C; also shown in a number of other fuxianhuids), but no similar structures are known from radiodont frontal appendages.
- (e) Kink: Radiodont frontal appendages bear their kink at the joint between shaft and podomere 1 of distal articulated region. On *L. ferox* appendage, the kink is situated between podomeres 3 to 5 (Figure 6B). The position of kink in *L. ferox* matches that in other fuxianhuid legs, suggesting similar functional morphology (e.g., *Chengjiangocaris longiformis*, Figs. 13, 14 in Hou and Bergström, 1997).
- (f) Number of podomeres: The number of podomeres in *L. ferox* appendage (11 + claw) falls within the known range of podomere numbers in fuxianhuid legs.
- (g) Arthroal membranes: Although not emphasized in the literature, the triangular shape of arthroal membranes is actually present in the legs of some fuxianhuids (e.g., Fig. 13 in Hou and Bergström, 1997; Fig. 16.5 b in Hou et al., 2004; Fig. 4C in Chen et al., 2020; Fig. 2E1 in Wu et al., 2019).
- (h) Podomere shape: The homonomous rectangular podomeres are common in fuxianhuids, whilst the podomeres of most radiodont frontal appendages are more trapezoid.
- (i) Endite: Most radiodont frontal appendages possess auxiliary spines on their endites, but auxiliary spines are absent in *L. ferox*. The endites of *L. ferox* (Figures 6B, C, 7B) are very similar to those in *Alacaris?* sp. (Figures 4, 5B).
- (j) Claw: The claw of *L. ferox* is certainly peculiar among the Cambrian euarthropods, but it has no counterparts in radiodonts.

Based on these points above, I insist that the *L. ferox* appendage is very likely a post-SPA leg of fuxianhuiid, and that the radiodont-like features of *L. ferox* appendage suggested in the manuscript are superficial. I would like to invite the authors to consider the information above, to incorporate useful points into the discussion of comparative morphology in the section ‘Affinities of *Lihuacaris ferox*’, and to revise their evolutionary discussion accordingly. The notion of *L. ferox* appendage as radiodont-like may be weakened throughout the text. I would respect if the authors keep their discussion on the segmental affinities of *L. ferox* appendage.

We thank the reviewer for providing this detailed discussion on the morphology of *Lihuacaris*. Although we would opt to maintain the precise affinities of this taxon as *incertae sedis* in order to be conservative in our interpretation, we have included an additional paragraph that discusses *Lihuacaris* as potential fuxianhuiid endopods, incorporating several of the observations made by the reviewer. We do note that various of the characters discussed by the reviewer are also found in radiodont frontal appendages, and have been discussed accordingly (e.g. length, shape of the podomeres, presence of triangular arthrodistal membranes, putative extensor muscle). Although we maintain the position that it is not possible to definitely conclude that *Lihuacaris* indeed belongs to either a radiodont or fuxianhuiid until more complete material is found, we hope that this will provide the reader with a balanced perspective on the enigmatic affinities of this organism.

Specific points:

Line 53. There is no lobopodian named as Macrodictyon. I suppose the authors might mean Megadictyon.

We thank the reviewer for pointing out this typo and have corrected it.

Lines 65-66. Aria et al. (2021, Journal of Geological Society) seems to be a better reference here for the post-tritocerebral hypothesis of fuxianhuiid specialized post-antennal appendages.

We have added this reference.

Lines 281-282. See point (b) above for the discussion on the length of appendage.

Lines 282-286. See point (h) above for the discussion on the shape of podomere.

Lines 286-289. See point (i) above for the discussion on the endite.

Lines 289-292. See point (d) above for the discussion on the linear groove or putative extensor muscle.

Lines 295-374. I would like to invite the authors to revise these paragraphs by considering the potentially helpful points (a-j) above.

Lines 376-437. The authors may revise this section ‘Position of *Lihuacaris ferox* in the euarthropod stem lineage’ if they change their ideas on the affinity of *Lihuacaris ferox*.

Lines 489-490. The conclusion on the possible affinities of *L. ferox* may require revision, as its appendage shows many fuxianhuiid features.

We have included an additional paragraph discussing the possible links with fuxianhuiids based on these recommendations, and also made changes the rest of manuscript accordingly to reflect this new discussion.

Figure 1B. The segmental affinity of radiodont frontal appendages was coded as ‘questionable’ (‘?’; protocerebral or deutocerebral) in the morphological matrix of Zeng et al. (2020). However, regardless of the uncertainty pointed out in the original reference, the illustration in panel B simply locates the frontal appendage at a protocerebral position. A correction recognizing the uncertainty needs to be made on panel B.

While the reviewer is correct, that Zeng et al. (2020) did code the segmental affinity of radiodont frontal appendages as questionable, in our view this does not accurately convey the position of the authors of that study. The authors summarize the evidence in support of a protocerebral innervation of radiodont appendages, and deutocerebral for megacheirans. They then offer the following explanation: ‘a straightforward explanation for this segmental mismatch would be a protocerebral to deutocerebral positional transformation of the frontalmost appendages during the evolution from Radiodonta to Kylinxia/Deuteropoda.’ Zeng et al. (2020, supplementary discussion). Thus we choose to keep figure 1B in its original form.

Figure 6. The abbreviations for tergite do not match in the panels (‘te’) and caption (‘t’). The explanation for abbreviation ‘gn’ is missing in the caption.

We thank the reviewer for spotting these errors, and have corrected them.

References:

Aria C., Zhao F., Zhu M. 2021 Fuxianhuidids are mandibulates and share affinities with total-group Myriapoda. *Journal of the Geological Society*, jgs2020-2246. (doi:10.1144/jgs2020-246).

Chen H., Legg D.A., Zhai D., Liu Y., Hou X. 2020 New data on the anatomy of fuxianhuid arthropod *Guangweicaris spinatus* from the lower Cambrian Guanshan Biota, Yunnan, China. *Acta Palaeontologica Polonica* 65(1), 139-148.

Hou X., Bergström J. 1997 Arthropods of the lower Cambrian Chengjiang fauna, southwest China. *Fossils and Strata* 45, 1-116.

Hou X., Aldridge R.J., Bergström J., Siveter D.J., Siveter D.J., Feng X. 2004 *The Cambrian fossils of Chengjiang, China: the flowering of early animal life*, Blackwell Science Ltd; 233 p.

Wu Y., Liu J. 2019 Anatomy and relationships of the fuxianhuid euarthropod *Guangweicaris* from the early Cambrian Guanshan Biota in Kunming, Yunnan, Southwest China revisited. *Acta Palaeontologica Polonica* 64(3), 543-548.

Zeng H., Zhao F., Niu K., Zhu M., Huang D. 2020 An early Cambrian euarthropod with radiodont-like raptorial appendages. *Nature* 588(7836), 101-105. (doi:10.1038/s41586-020-2883-7).

Reviewer: 2

Comments to the Author(s)

Comments:

This manuscript by Jiao and his colleagues described two new arthropods, *Lihuacaris ferox* gen. et sp. nov. and *Alacaris?* sp. from Cambrian Guanshan deposit, in combination with a discussion on the evolution of euarthropod characters and the feeding strategy.

It is no doubt that *Lihuacaris ferox* represents an interesting new species, whose appendage shows unique combined features of Radiodonta and Deuteropoda. The morphology of the appendage, especially its unique spinose distal element, expanded the morphospace of the euarthropod limbs. I agree with authors that the new taxon most likely falls close to radiodonts in the Euarthropoda, though lack of the whole-body information precludes a precise systematic assignment within the Arthropoda. I am also happy to see various evolutionary scenarios suggested by authors based on the different positions of *Lihuacaris* within the total group of Euarthropoda.

We thank the reviewer for their positive comments, and support for our conclusions.

For the latter taxon, *Alacaris?* sp., the authors identified the specimens as isolated fuxianhuiid endopods, and assign into family Chengjiangacaridae. But I am hesitating to define *Alacaris?* sp. fossils as the endopod. The endite described in the manuscript, marked as “en” particular in Figure 4 and 5 show striking similarities with the pleural spine of the dorsal tergite, particular that in *Fuxianhuiida*. The size of structure sounds too big to be the endopod of fuxianhuiids. Because in the group of *Fuxianhuiida*, each dorsal tergite covers multiple appendages. However, the “height of the podomere” of the endopod is 7 mm (see Line 47), that is three times wider than the width of the tergite closely preserved to it (Fig. 4). Furthermore, the evidence of the teeth and slender spines attached to the hypothetical protopod is not very convincing. I found that the so called protopod part, preserved on the upper bedding and covered some segments (see Figs. 3 and 4), is much more similar to the dorsal exoskeleton of euarthropod.

I have not seen the fossils in person, so it has the possibility of being the endopod. So, it is strongly recommended a double check of *Alacaris?* sp. specimens, just in case. But the importance of this manuscript won't be affected by the definition of the endopod in *Alacaris?* sp.

We thank the reviewer for offering this possible alternative interpretation for these specimens.

We understand their point of view, however we argue that the following observations support the interpretation of these specimens as fuxianhuiid endopods rather than as a dorsal exoskeleton of a euarthropod.

a) the gnathobase – the presence of a gnathobase is hard to reconcile if the endopod is instead considered a dorsal exoskeleton.

b) we do not suggest that the tergite in Fig. 4 is part of the same animal as the endopod, indeed we argue the opposite. If instead the specimen is a dorsal exoskeleton (not an endopod), it is still preserved close to an unrelated tergite.

c) slender spines are preserved in multiple specimens (see Fig. 4 and Fig. 5).

In addition, I would suggest to use “prodopod” instead of “prodopodite” when describe a non-crustacean arthropod.

We follow the terminology of Boxshall (2004), and use the term ‘protopodite’ to refer to the proximal part of the biramous limb that carries the rami. We have added a sentence that clarifies this point to the Terminology section.

Some minor mistakes:

Line 63: should be [e.g. 18,19,20]

We thank the reviewer for spotting this error and have corrected it.

Line 134-149: in this part, the number of the specimens described in the text was mismatched with the specimen number in the Figures 3-5. For example, in line 136, the authors mentioned the specimen 12434 (Fig. 4), while the specimen number is 12435 in the figure caption. Please check this part.

We have corrected the numbering in this section.

Line 201: add space between et and sp.

Corrected, thank you.

In summary, it is a well written manuscript. I recommend publication in Royal Society Open Science after the minor revision.

Reviewer: 3

Comments to the Author(s)

The authors describe in this paper two new species of Euarthropoda, one of them being the youngest occurrence of Chenjiangocaridae. The material is interesting in the context of arthropod evolution and merits publication. The text and the figures are in general of good quality, but I have some comments on some language inconsistencies and on improvements/amendments to the figures (see below). I recommend publication after minor revision.

20: diverse can be used in different ways, better use species-rich.

Changed, thank you.

20: As the rank height for a group is rather arbitrary, “the most diverse animal phylum” has no true meaning in a scientific way. I suggest to rephrase the sentence.

Phyla are well established and well known groupings of organisms. This sentence is intended as an introductory statement, and we choose to keep it as written as it conveys the message that we are aiming for.

22: Definitions are done in humanities, I suggest “characterise” instead.

22: Though often used, “body plan” implies a planner, which is problematic in the days of Intelligent Design and other creationists, better rephrase.

We have rephrased this sentence to reflect the two points above.

26: here and in other places: try to avoid Linnean ranks (e.g. also in the Systematic palaeontology section)

Linnean ranks can be useful, so we have chosen to keep them in multiple places in the manuscript, though we have removed a few (as suggested below) where we think it aids readability.

28: here and in other places: Use also the species name if you are not explicitly talking about the genus only.

Changed throughout the abstract.

29: “a hypertrophied”, not an, as the h is voiced

Changed. Thank you.

50 ff.: Sentence starting with “Among” is erroneous grammatically, please check.

In this context it is ok to start the sentence with ‘Among’.

56: Better phrase “modern representatives of Euarthropoda”

Rephrased, thank you for the suggestion.

56: As synapomorphies are new characters shared by two groups, the sentence is phrased a bit unfortunate. If these characters would be autapomorphies of Euarthropoda, that would not fit with their occurrence also in radiodonts. Most probably, these characters already occurred earlier and are hence plesiomorphies, but characteristic for Euarthropoda (and radiodonts). Please rephrase.

This sentence has been rephrased, ‘synapomorphies’ replaced with ‘morphological features’.

171: Here it becomes especially apparent that deleting “family” would be easy and even improve the readability.

Deleted the word ‘family’.

175: Here, taxon can be changed to species, which provides more information (check also in other places).

Changed here.

210: caris is not Latin, but Greek

The Latin word ‘caris’ also translates approximately to ‘crab’ or ‘shrimp’. We have added a reference to the original Greek origin to the etymology sentence.

273: I recommend to restrict segment to body segments, for appendages it is better to use elements (check also in other places).

‘Elements’ does not quite provide the meaning that we require for these statements. We choose to retain the use of ‘segments’. It is clear from the context of our sentences that we are referring to appendages, and not body segments, so there should not be confusion.

278: Radiodonta as group name should be used as singular (though often plural is used by many authors).

We have replaced ‘Radiodonta’ with ‘radiodonts’.

322: “a hypertrophied”, not an, as the h is voiced

Corrected. Thank you.

350: typo in Lihuacaris

Corrected. Thank you.

384: Lihuacaris should not be italicised as the rest of the header is (also check other headers)

Corrected – in this header and the following ones.

388 ff.: The function of these appendages does not depend on their segmental affiliation, but the sentence creates this impression, please rephrase.

This sentence refers to the function of the ‘base’ of *Lihuacaris ferox* appendages. Having the same segmental affinity and position on the head as radiodont appendages, the function of the ‘base’ and ‘shaft’ would be similar. We have rephrased this sentence to make this meaning explicit and to avoid creating the wrong impression.

404/405: The appendages for sure do not arise from the deutocerebrum, but they may have been innervated by the deutocerebrum or may arise from the deutocerebral segment, please rephrase.

Changed, now refers to the deutocerebral segment.

405: Not Lihuacaris is similar to the first appendages, but its appendages are similar. Be careful with the references.

Thank you for pointing out this error, we have corrected it. We have checked the references in this sentence (*Kiirsortoquia* – Stein 2010; *Bushizheia* – O’Flynn et al. 2020).

421 ff: Sentence incomplete, please check.

We have added the word ‘as’ to complete the sentence.

833: typo in megacheirans

Corrected, thank you.

Fig. 1: The appendages should all be positioned in the same orientation (e.g. spines to the bottom) to allow better comparability.

The appendages have been displayed in different orientations to reflect the differences between radiodonts and *Kylinxia*/ megacheirans. For the former, we orientate spines to the bottom, for the latter spines upwards – this reflects the orientation of the appendages and how they insert into the head.

Fig. 2. The font is in most cases too small, hardly readable (please check also journal guidelines). Better rearrange the figure to more square-shape, then you can make everything a bit larger.

We thank the referee for their suggested changes to Figure 2, and have provided an updated (and more readable) figure.

Figs. 3–7: For future photographs I recommend to use cross-polarised light (polarisation filter in front of camera lens, perpendicular polarisation filters in front of light source) as that strongly enhances the contrast between fossils and matrix. For the current photos, I recommend to optimise the histogram for a better contrast.

We thank the reviewer for their suggestions. We edited the color levels in the photographs before our first submission. We feel that the outlines of the specimens are all clearly visible. We are happy to work with the reviewer and editors if there are specific features they feel are not visible.

Fig. 8: There is a reconstruction of *Lihuacaris ferox*, but I would also like to see one of *Alacaris* ?, even if incomplete.

We do not feel that a reconstruction of *Alacaris?* is warranted at this time. The information available in the figures, especially Figure 4, provide as much detail as is possible at the moment. The reconstruction of *Lihuacaris ferox* is supplied to facilitate comparison with radiodonts, stomatopods, and artiopodans, whereas a reconstruction of *Alacaris?* would not serve such a purpose.

Fig. 9: You need to explain what the grey lines in the phylograms represent. I also think that it would be necessary to name the presumed outgroup in B to be able to place the apomorphy “arthropodized limbs” at that position.

We have added a sentence to the caption to explain what the grey lines represent. The numbers in B indicate possible positions for *Lihuacaris ferox*. If *L. ferox* occupies the position indicated by the ‘1’, then it is the earliest diverging euarthropod with arthropodized limbs, hence why the apomorphy is positioned there.

Carolin Haug, LMU Munich

Reviewer: 4

Comments to the Author(s)

This is a good MS based on very accurate descriptions of a fragmentary but well-preserved arthropod material from the Guanshan Lagerstätte, with interesting discussions on euarthropod early evolution and autecology. It is well written and easy to follow. I would suggest to slightly re-arrange/shorten the paragraph on the assumed feeding mode of *Lihuacaris* (see below). Figures and diagrams are of excellent quality. Figure 1 is particularly useful to those readers who may not be familiar with current debates on euarthropod evolution. I would be happy to recommend this MS for publication in this journal.

We thank the reviewer for their positive comments and helpful suggestions for improving our manuscript. It is particularly nice to hear that Figure 1 will be useful for a wide audience.

Line 88- Please give some general information on the faunal composition of the Guanshan Biota and especially how it differs from other Lagerstätten (e.g. Chengjiang). Why do you think that shallower environments resulted in radiodont endemicity? Do we know the approximate age of this formation (e.g. biozonations)?

We are aware that the manuscript is already quite long, and so we aimed to keep this part of the text as short as possible. We have added in a few additional details to this paragraph, but stop short of adding so much that the manuscript becomes unwieldy. A recent paper (Jiao et al. 2021) discusses the radiodonts and their endemicity in detail, so we do not repeat those arguments here. The approximate age of the levels of interest (*Palaeolenus* to *Megapaleolenus* trilobite zones, Canglangpuan regional stage) is given in the text.

Line 125 +

The description of *Alacaris?* sp. is based on two isolated appendages. I am just wondering whether more abundant fragments of other body features could be found in the Guanshan Biota, that would more strongly support its affiliation to *Alacaris?* Is this biota characterized by disarticulated arthropod fragments?

Since your figures look quite large, why not figure a complete specimen of *Alacaris* (from a different locality) in one of them. That would help readers locate the isolated appendages shown in the MS. Perhaps ask a colleague to provide such illustration?

Line 165: again here (see above) a picture of *Guangweicaris* would be useful.

There are numerous disarticulated gnathobases and tergites (for example those in Figure 4 and Figure 6) in the material studied. However, we do not feel that it would be appropriate to associate these with the endopods of *Alacaris?* Firstly, those on the same slab in proximity to *Alacaris?* endopods are too small (unless they represent reduced anterior tergites), as noted in the text. Secondly, other specimens are associated with bivalve euarthropod carapaces (see Figure 5A). Thus, as it is clear that fragments of numerous different animals have been deposited together, we cannot at this time combine evidence from different fragments to support our interpretation of the endopods as belonging to a new species of *Alacaris*.

We also do not want to add more figures at this stage (we already have 9 figures) – *Guangweicaris* and *Alacaris* have both been recently figured in open access publications (see, for example, Chen et al. 2020 for *Guangweicaris* and Yang et al. 2018 for *Alacaris*)

Line 169: Perhaps what you have here is the isolated appendages of large specimen of *Guangweicaris* (see number of podomeres). It would not be surprising that large specimens had more chance to be disarticulated by mudflows or currents.

We outline in this section why we do not think that these appendages belong to *Guangweicaris*. The number of podomeres is too high in our new specimens, and a gnathobasic protopodite is only known in *Alacaris*.

Does *Xiaocaris* really lack gnathobasic protopodites? Perhaps they are concealed by other features, Please check.

Line 177- That's why I made the remark above.

Knowledge of the walking limbs of *Xiaocaris* comes from CT data (Liu et al. 2020). Thus, because the fossils have been visualised in three dimensions, we can be confident that *Xiaocaris* truly lacks gnathobasic protopodites.

Line 186- Please, be careful here. Can you estimate the approx. size of the arthropod that bore the isolated appendage? Comparisons should be based on specimens with a comparable size.

We are unsure what this comment refers to, as we make no mention of size estimates based on the isolated appendage.

Line 201. *Lihuacaris ferox*

This new species is mainly based on two isolated appendages (one of them being incomplete). I agree that at least one appendage provides clear morphological information. However, isn't it a bit risky to make a new genus and species with such a fragmentary material?

The isolated appendages provide diagnostic characters that allow the new species to be identified. Other Cambrian euarthropods have been diagnosed based on isolated appendages, for example radiodonts from the Guanshan (e.g. Wang et al. 2013; Jiao et al. 2021).

Line 233-234

Not very clear here. Is it the carapace ornament of the bivalved arthropod?

No. This sentence refers to a different rectangular feature. We have reordered the sentences in this paragraph to make this clearer.

Line 237

I am not sure that the holotype of a new species can be an isolated appendage. It would mean that this new taxon is defined of a very small percentage of its whole anatomy. We don't even know the size of the whole animal. Please check.

As outlined above, the isolated appendages provide diagnostic characters that allow a new species to be diagnosed. Other Cambrian euarthropods have been diagnosed based on isolated appendages.

Line 281

It seems that other radiodonts occur in the same biota. Detailed comparisons should be made with them (again images would be useful).

Guanshan radiodonts have been described by Wang et al. (2013), and more recently Jiao et al. (2021). We propose that *Lihuacaris ferox* is not a radiodont, but instead closely related to radiodonts. We provide a detailed comparison between *Lihuacaris ferox* and relevant radiodonts, not limiting ourselves to those described from Guanshan.

Line 290

Do you have muscle remains?

We describe a linear feature that runs through the appendage, which is tentatively described as a possible extensor muscle (labeled in the figures).

Line 307

It is clear that this appendage resembles that of numerous radiodont species. What makes it so special is not completely clear to me. What do you mean by the “level of separation”. This character may result from post-mortem decay of muscles

We agree that in some ways *Lihuacaris* resembles some radiodonts, as we discuss in the main text. However, we also discuss in detail how *Lihuacaris* differs from radiodonts, for example in the lack of dorsal spines and the large, multispined, distal element. We have removed the sentence referring to the level of separation between base and proximal podomere, as we want the focus to rest on these other, more substantial, differences.

Line 318

Do you mean that the appendages of *Lihuacaris* had a special kind of highly flexible articulation?

No. This part compares *Caryosyntrips* to *Lihuacaris*, specifically the whole appendage of the former with the distal element of the latter. *Lihuacaris* displays podomeres separated by membranes proximal to its distal element. In this section we explain that the proximal part of *Caryosyntrips* appendages is very different to what is observed in *Lihuacaris*. In *Lihuacaris* the multiple articulating podmeres would allow much more flexibility than the single bell-shaped podomere of *Caryosyntrips*.

Line 324

There is no speculation here. the terminal “claw” looks like a fused element (no podomere boundaries visible)

The speculation would be assuming a radiodont affinity for *Lihuacaris* – we do not have enough evidence to support this hypothesis. We have added the word ‘radiodont’ before terminal claw, to make this explicit.

Line 333

Yes, interesting hypothesis.

Thank you.

Line 384

Yes, it is one option to be discussed. However, since your appendages are isolated, we don’t know to which part of the head they were attached. I find this part (>> 438) a bit speculative?

Line 402

Same here. One of the three options is correct but additional fossil evidence (relation to head) critical to the discussion is lacking.

These subsections run through the implications for different segmental affinities for *Lihuacaris* appendages. We choose to keep these sections as they emphasise the possible importance of *Lihuacaris*. We agree that the fossil evidence is lacking and that is why we discuss all the possibilities. This allows us to emphasise the importance of future discoveries of *Lihuacaris* for resolving outstanding issues.

Line 441

These appendages are quite large and bear a strong terminal “claws”. Is it enough to make them predators? I see no obvious prehensile features unless the appendage. *Sidneyia* could maintain small prey along its ventral part and masticate them by using strong gnathobases. I see no such hard elements in (e.g.) Fig. 8.

We hypothesise that the hooked part of the distal element is used to capture prey. These appendage could then flex, as demonstrated by the presence of triangular membrane, to pass prey to the mouth. Without the rest of the body, this part is still tentative, however we prefer to leave in this discussion, which will be augmented by future discoveries.

Line 444

Comparisons with mantis-shrimps seems to be a bit far-fetched. I don't find them very convincing. I see no raptorial elements in the appendage of *Lihucaris* unless they could fold up ventrally. If possible provide a diagram (add to Fig. 8 ?) showing both the "resting" and "folded" position. You could do the same for the other appendages shown in Fig. 8. It would greatly help readers understand the function of appendages.

We do not suggest that *Lihucaris* used its distal element in the same way as mantis shrimp use their dactyl, nor do we suggest that the raptorial elements in the *Lihucaris* appendage could fold up ventrally. We agree that *Lihucaris* likely did not feed like a mantis shrimp. However, because of the striking similarities between the distal element of *Lihucaris* and the dactyl of spearing mantis shrimp, we feel the need to include the comparison in the discussion.

Line 452

Sidneyia and stomatopods are two different types of animal in terms of predation. *Sidneyia* was most probably a slow animal feeding on epibenthic prey. Mantis shrimps are extremely fast hunters.

We discuss *Sidneyia* and stomatopods in order to contrast them with *Lihucaris* and conclude that there is not strong evidence for *Lihucaris* being a durophagous predator. We agree that stomatopods and *Sidneyia* are two different types of animal in terms of predation – this is of importance as we show that *Lihucaris* is different to both, despite similarities in some parts of the appendage to both stomatopods and arthropods.

Line 452 +

You are embarking into detailed discussions on the feeding mode of *Lihucaris*. Think that these discussions are based on a couple of isolated appendages only. It might be a bit risky to discuss the feeding mode of an animal on the basis of such a fragmentary material. We know nothing about its size, morphology of other appendages, gut, etc... Please improve this part.

We agree that more fossil evidence is required in order to better understand the feeding mode of *Lihucaris*. We retain this paragraph in the manuscript because some discussion, albeit tentative, is warranted of the ecology of this animal.

Line 464. You see... (impossible)

Line 465. Idem

Perhaps start this paragraph by saying that unfortunately very little is known of the actual functional morphology of the animal.

We have added a statement referring to the limits of what can be inferred from isolated appendages of uncertain segmental affinity to the start of this paragraph.

Line 468

Yes, I find it interesting to mention the feeding habits of *Fuxianhuia*. This is a well- documented taxon with plenty of excellently preserved specimens. Unfortunately again, your specimens are fragmentary. It is frustrating in some way. Perhaps, try to avoid "what-if-they-had" long discussions.

Although fragmentary, the specimens of *Alacaris*? described in our contribution display gnathobases and endites. Comparison with other fuxianhuuids allows a limited discussion of the

feeding habit of this new taxon. Of course, more fossil material of more complete specimens will allow further refinement, but we can still offer some insights based on our fragmentary material. We do not feel that the seven lines is excessive, and we include clear limits on what our material allows us to say about the feeding of this fuxianhuiid.

Line 483

This statement is perfectly correct.

Although the present material is fragmentary, new studies of the Guanshan Biota might reveal unknown aspects of arthropod evolution. Try to stress on the importance of this biota (its potential interest, environmental context, etc.)- Also in the abstract.

Thank you.

Jean Vannier

References cited in response to referees

- Boxshall, G.A., 2004. The evolution of arthropod limbs. *Biological Reviews*, 79(2), pp.253-300.
- Chen, H., Legg, D.A., Zhai, D.Y., Liu, Y. and Hou, X.G., 2020. New data on the anatomy of fuxianhuiid arthropod *Guangweicaris spinatus* from the lower Cambrian Guanshan Biota, Yunnan, China. *Acta Palaeontologica Polonica*, 65(1), pp.139-148.
- Jiao, D.G., Pates, S., Lerosey-Aubril, R., Ortega-Hernández, J., Yang, J., Lan, T. and Zhang, X.G., 2021. The endemic radiodonts of the Cambrian Stage 4 Guanshan Biota of South China. *Acta Palaeontologica Polonica*, 66(2), pp.255-274.
- Liu, Y., Ortega-Hernández, J., Chen, H., Mai, H., Zhai, D. and Hou, X., 2020. Computed tomography sheds new light on the affinities of the enigmatic euarthropod *Jianshanina furcatus* from the early Cambrian Chengjiang biota. *BMC Evolutionary Biology*, 20, pp.1-17.
- Yang, J., Ortega-Hernández, J., Legg, D.A., Lan, T., Hou, J.B. and Zhang, X.G., 2018. Early Cambrian fuxianhuiids from China reveal origin of the gnathobasic protopodite in euarthropods. *Nature Communications*, 9(1), pp.1-9.
- Wang, Y., Huang, D. and Hu, S., 2013. New anomalocardid frontal appendages from the Guanshan biota, eastern Yunnan. *Chinese Science Bulletin*, 58(32), pp.3937-3942.